# Evolution of connectivity architecture in the Drosophila mushroom body

Kaitlyn Elizabeth Ellis[1], Sven Bervoets [1], Hayley Smihula [1], Ishani Ganguly [2], Eva Vigato [1], Thomas O. Auer [3,4], Richard Benton [3], Ashok Litwin-Kumar[2] & Sophie Jeanne Cécile Caron [1]✉

Brain evolution has primarily been studied at the macroscopic level by comparing the relative size of homologous brain centers between species. How neuronal circuits change at the cellular level over evolutionary time remains largely unanswered. Here, using a phylogenetically informed framework, we compare the olfactory circuits of three closely related *Drosophila* species that differ in their chemical ecology: the generalists *Drosophila melanogaster* and *Drosophila simulans* and *Drosophila sechellia* that specializes on ripe noni fruit. We examine a central part of the olfactory circuit that, to our knowledge, has not been investigated in these species—the connections between projection neurons and the Kenyon cells of the mushroom body—and identify species-specific connectivity patterns. We found that neurons encoding food odors connect more frequently with Kenyon cells, giving rise to species-specific biases in connectivity. These species-specific connectivity differences reflect two distinct neuronal phenotypes: in the number of projection neurons or in the number of presynaptic boutons formed by individual projection neurons. Finally, behavioral analyses suggest that such increased connectivity enhances learning performance in an associative task. Our study shows how fine-grained aspects of connectivity architecture in an associative brain center can change during evolution to reflect the chemical ecology of a species.

Brain evolution has been primarily studied at the macroscopic level by comparing gross neuroanatomical features in homologous brain centers across distantly related species[1–3]. This pioneering work revealed that, over intermediate evolutionary timescales, the number of specialized brain centers does not change considerably, but the types and numbers of neurons forming these centers can vary greatly. Recent advances in comparative transcriptomics have provided new insights into the evolution and diversification of neurons, revealing how subtle variations in highly conserved regulatory gene networks can give rise to drastic changes in the rate at which neuronal progenitors proliferate or the types of neuron they give rise to[4]. To what degree such changes reflect selection pressures remains unclear. Moreover, how such

changes manifest themselves at the level of neuronal circuits is not yet understood, as it remains technically challenging to delineate neuronal circuits in non-traditional model systems and to compare them across species with different evolutionary trajectories[5].

Flies in the genus *Drosophila* have evolved to exploit a remarkable diversity of ecological niches[6]. The phylogeny of most *Drosophila* species has been resolved, revealing that even closely related *Drosophila* species can inhabit drastically different environments[7]. For instance, *Drosophila sechellia*, a species endemic to the Seychelles islands, is a specialist for noni—a toxic fruit that produces a distinctive bouquet of pungent acids—whereas its closest relative, *Drosophila simulans*, is a generalist and a human commensal that can be found in

[1]School of Biological Sciences, University of Utah, Salt Lake City, USA. [2]Center for Theoretical Neuroscience, Columbia University New York, USA. [3]Center for Integrative Genomics, Faculty of Biology and Medicine, University of Lausanne, Lausanne, Switzerland. [4]Present address: Department of Biology, University of Fribourg, Fribourg, Switzerland. ✉e-mail: sophie.caron@utah.edu

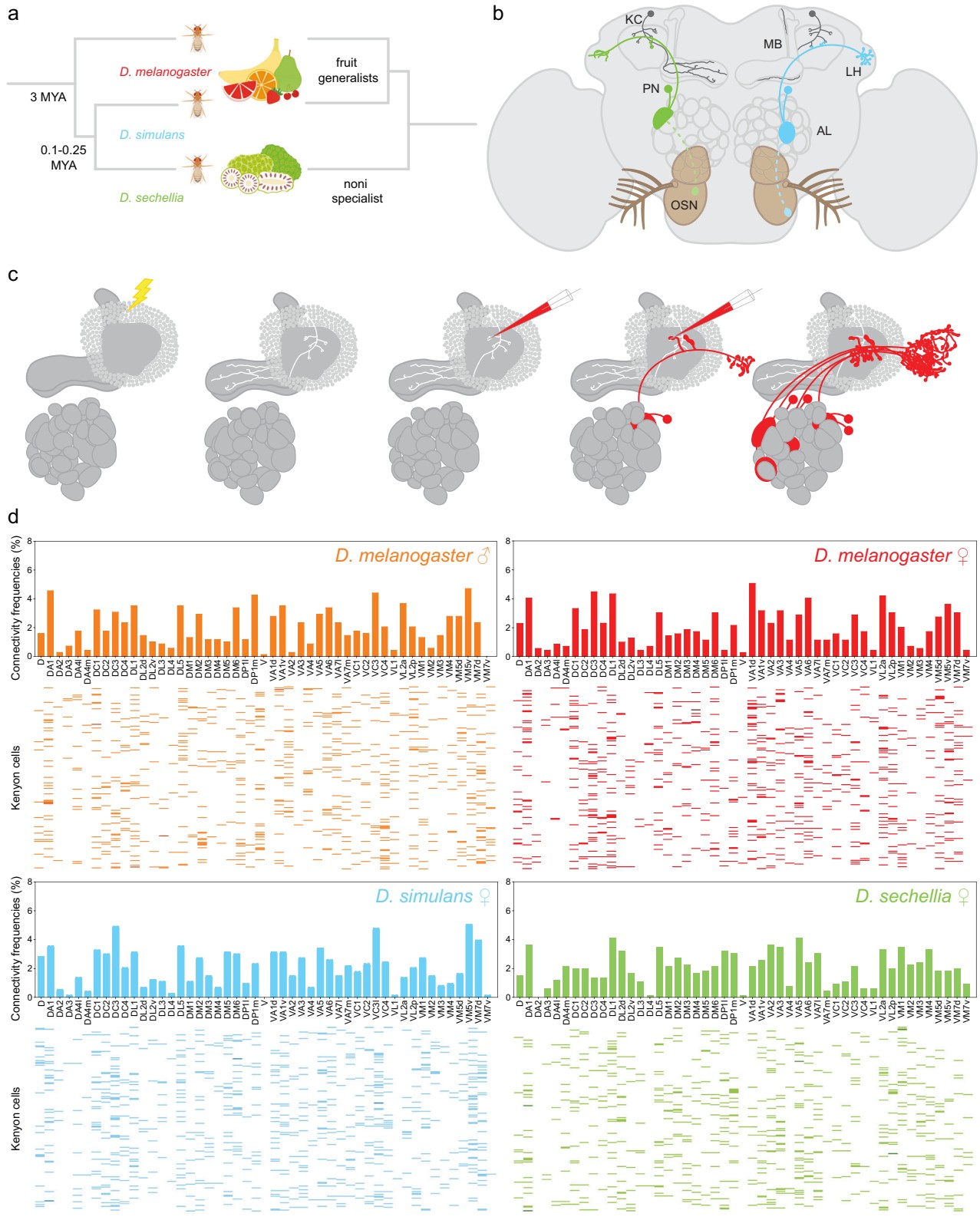

most cosmopolitan areas[8–11] (Fig. 1a). A mere 0.1-0.25 million years of evolution separate these two species from their common ancestor, and only 3 million years separate their common ancestor from their well-studied relative, *Drosophila melanogaster*[7]. *D. melanogaster* is also a human commensal whose ecology largely overlaps with that of *D. simulans*[12]. These inverse relationships–phylogenetically, *D. sechellia* is closer to *D. simulans* while ecologically *D. melanogaster* and *D. simulans* are more alike–make this group of *Drosophila* highly suitable for investigating how neuronal circuits evolve over comparatively short timescales.

Olfactory driven responses used to locate food sources are likely one of the most important ways whereby a species can adapt behaviors

**Fig. 1 | Mapping Kenyon cell inputs in *Drosophila* species living in different ecological niches. a** Schematic depicting the phylogenetic relationships of *D. melanogaster* (red), *D. simulans* (blue), and *D. sechellia* (green) on the left and their ecological relationships on the right. **b** Schematic depicting the *Drosophila* olfactory circuit: olfactory sensory neurons that express the same receptor gene(s) (OSNs, green and blue neurons with dotted outline) converge onto the same glomerulus in the antennal lobe (AL); projection neurons (PNs, green and blue neurons with full outline) connect individual glomeruli to the mushroom body (MB) and the lateral horn (LH); Kenyon cells (dark gray) receive input from a small number of projection neurons. **c** Simplified schematic depicting the technique used to map connections between projection neurons and Kenyon cells: a Kenyon cell is photo-labeled (white) and the projection neurons connected to each of its claw are dye-labeled (red) such that the antennal lobe glomeruli innervated by the labeled projection neurons can be identified; see Supplementary Fig. 3 for a more detailed description of the technique. **d** Connections between glomeruli and Kenyon cells

were mapped in *D. melanogaster*, *D. simulans* and *D. sechellia*, and all connections are reported in four connectivity matrices (*D. melanogaster* males: top left panel and orange (687 connections); *D. melanogaster* females: top right panel and red (704 connections); *D. simulans* females: bottom left panel and blue (717 connections); *D. sechellia* females: bottom right panel and green (692 connections)). In each matrix, a row corresponds to a Kenyon cell—there are 200 Kenyon cells per matrix—and each column corresponds to the different antennal lobe glomeruli; each colored bar indicates the input connections of a given Kenyon cell, and the intensity of the color denotes the number of connections found between a particular Kenyon cell and a given glomerulus (light: one connection; medium: two connections; dark: three connections). The bar graphs above the matrices represent the frequencies at which a particular glomerulus was connected to Kenyon cells as measured in a given matrix. All source data used in this figure are provided in the Source Data file.

to its ecological niche. In *D. melanogaster*, the majority of olfactory sensory neurons express only one receptor gene, and the axons of neurons expressing the same receptor gene(s) converge on a specific glomerulus in the antennal lobe, forming a stereotypical map[13,14] (Fig. 1b). The genomes of *D. melanogaster*, *D. simulans* and *D. sechellia* harbor a comparable number of functional olfactory receptor genes, and homologous glomeruli can be identified in the antennal lobe in these species[15–17]. In *D. sechellia*, however, a few olfactory sensory neurons show species-specific responses to noni odors: Or22a-expressing neurons are preferentially activated by methyl esters whereas Ir75b-expressing neurons are most sensitive to hexanoic acid[16,18–21]. By contrast, in *D. melanogaster*, Or22a-expressing neurons are broadly tuned to different ethyl esters whereas Ir75b-expressing neurons are broadly tuned to shorter-chain acids[16,18–21]. These species-specific tuning properties are due to specific amino acid differences in the presumed ligand-binding domain of Or22a and Ir75b receptors[16,21]. In addition, there are two- to three-fold more Or22a- and Ir75b-expressing neurons in *D. sechellia*, and the glomeruli innervated by these neurons, DM2 and DL2d, respectively, are larger[16,18,19]. Other glomeruli—such as VM5d, which is innervated by Or85c/b-expressing neurons—are also larger in *D. sechellia* compared to *D. melanogaster* due to increased sensory neuron numbers[16,19,20].

Whether and how the changes that occurred at the levels of receptor proteins and sensory neurons are reflected downstream, at the level of higher processing centers, is largely unknown.

Projection neurons innervating individual antennal lobe glomeruli transmit olfactory information to the mushroom body, an associative brain center, and the lateral horn, a center that mediates innate responses to odors[22] (Fig. 1b). The projection neurons of the antennal lobe and the Kenyon cells of the mushroom body have been studied in detail in *D. melanogaster*. Each glomerulus type is innervated by a distinct, but largely stereotyped number of projection neurons, from one up to eight[23]. The mushroom body consists of about 2000 neurons, called Kenyon cells, that can be divided into three major types (α/β, α′/β′ and γ Kenyon cells); each Kenyon cell receives input from a small number of projection neurons, on average seven[24,25]. The connectivity architecture between projection neurons and Kenyon cells has been resolved, revealing two important principles: first, these connections are unstructured, in that individual Kenyon cells integrate inputs from a random set of projection neurons; second, some types of projection neuron connect more frequently to Kenyon cells than others, leading to a biased representation of glomeruli in the mushroom body[25–28]. These two connectivity patterns—randomization of input and biased connectivity—are genetically hardwired, suggesting that they might be shaped by selection pressures[29].

Here, we show that randomization of sensory input and biased connectivity are conserved features of the mushroom body architecture, observed not only in *D. melanogaster* but also in related species such as *D. simulans* and *D. sechellia*. However, we found that the

identity of the most biased projection neurons, which are either underrepresented or overrepresented, varies across species that occupy different ecological niches. Specifically, projection neurons detecting fermenting food odors are prioritized in the human commensals *D. melanogaster* and *D. simulans*, whereas projection neurons detecting noni are prioritized in *D. sechellia*. Additionally, our findings reveal that these differences in connectivity are attributed to morphological differences of the projection neurons connecting individual glomeruli to the mushroom body, either through changes in the number of projection neurons or through changes in the number of presynaptic boutons formed by individual neurons. Furthermore, our study suggests that increased connectivity to the mushroom body enhance learning performance in associative tasks.

## Results

### Randomization of input is conserved across *Drosophila* species

To compare features of the connectivity architecture between projection neurons and Kenyon cells in *D. melanogaster*, *D. simulans* and *D. sechellia*, we first resolved gross anatomical features of the antennal lobes of these species. Using confocal images of brains immunostained with a neuropil marker, we reconstructed entire antennal lobes by manually tracing the borders of individual glomeruli on single planes and projecting their volumes in three-dimensional space (Supplementary Fig. 1, Supplementary Data 1). The resulting projections preserved the shape, volume and location of all the glomeruli forming an antennal lobe. Each glomerulus was annotated using the well-characterized *D. melanogaster* antennal lobe map, which was generated using similar methods, as a reference[13,14,30]. Glomerular volumes were compared across species (Supplementary Data 1). We found that the antennal lobe map is largely conserved in the three species and contains a total of 51 glomeruli that can be recognized based on their shape and location. The volumes of most glomeruli are comparable across species with a few exceptions, including the noni-responsive DL2d, DM2 and VM5d glomeruli, which are larger in *D. sechellia*, as previously reported (DL2d: *D. melanogaster*: $1565 \pm 334\,\mu m^3$ ($n = 3$); *D. simulans*: $2007 \pm 185\,\mu m^3$ ($n = 3$); *D. sechellia*: $3066 \pm 296\,\mu m^3$ ($n = 3$); DM2: *D. melanogaster*: $3139 \pm 227\,\mu m^3$ ($n = 3$); *D. simulans*: $3169 \pm 227\,\mu m^3$ ($n = 3$); *D. sechellia*: $4594 \pm 127\,\mu m^3$ ($n = 3$); VM5d: *D. melanogaster*: $905 \pm 115\,\mu m^3$ ($n = 3$); *D. simulans*: $1773 \pm 127\,\mu m^3$ ($n = 3$); *D. sechellia*: $3793 \pm 153\,\mu m^3$ ($n = 3$))[16,17,19,21]. Despite these volume differences, the antennal lobes of the three species investigated are macroscopically nearly identical, which is in line with the notion that gross anatomy is conserved over the fairly short evolutionary distances separating the three species.

We next set out to compare the global connectivity architecture of the mushroom body across the three species by adapting the technique we previously developed to map projection neuron–Kenyon cell connections in *D. melanogaster*[26] (Fig. 1c). For each species, individual Kenyon cells were photo-labeled in flies

carrying the broad neuronal driver *nSynaptobrevin-GAL4* and a *UAS-photoactivatable-GFP* effector transgene. In *D. melanogaster*, α/β, α'/β' and γ Kenyon form a variable number of claw-shaped dendritic terminals. α/β, α'/β' and γ Kenyon cells were found in all three species, appear morphologically indistinguishable from one another and form on average a comparable number of claw-shaped dendritic terminals[24] (Supplementary Fig. 2). To identify the projection neurons connected to a photo-labeled Kenyon cell, a red-dye was electroporated sequentially in most of the claw-shaped dendritic terminals formed by that cell (Fig. 1c, Supplementary Fig. 3). Using this technique, the inputs of hundreds of Kenyon cells were identified in terms of the glomeruli from which they originate. We reported these results in a connectivity matrix that summarizes the glomerular inputs to 200 Kenyon cells. Statistical analyses of the resulting matrix can be used to reveal structured patterns of connectivity, such as whether groups of glomeruli are preferentially connected to the same Kenyon cells or whether projection neuron–Kenyon cell connections are random and biased[26,29]. We generated a total of four such connectivity matrices (Fig. 1d): two using *D. melanogaster* males and females (a total of 687 and 704 connections, respectively), one using *D. simulans* females (717 connections) and one using *D. sechellia* females (692 connections).

We performed an unbiased search for potential structural features in these connectivity matrices using principal component analysis. The variance associated with individual principal component projections provides a sensitive measure of structure as we have previously shown[26,29]. We extracted correlations within a given experimental connectivity matrix and compared them to correlations extracted from matrices in which connections were randomly shuffled. In these shuffled matrices, the total number of connections between projection neurons and Kenyon cells is the same as the number of connections reported in each experimental matrix, but the connections were randomly assigned (1000 uniform shuffle matrices). We also generated a second set of shuffle matrices in which the connections were scrambled but the frequencies at which projection neurons from each glomerulus connect to Kenyon cells were fixed to reflect the frequencies measured experimentally (1000 biased shuffle matrices). We found that the observed spectrum of variances is not significantly different to that of the biased shuffle matrices, but does significantly deviate from that of the uniform shuffle matrices (Fig. 2a, b). These results suggest that there are no detectable structural features in the experimental matrices other than the structure generated by the biases in connectivity frequencies.

## Biases in connectivity correlate with the chemical ecology of a species

If the connections between projection neurons and Kenyon cells were completely random, we would expect Kenyon cells to integrate input uniformly across glomeruli, and each glomerulus to have a connectivity frequency of about 2%. Yet, we found that the connections between projection neurons and Kenyon cells are biased: some glomeruli are overrepresented and have a connectivity frequency significantly higher than 2%, whereas some glomeruli are underrepresented and have a connectivity frequency significantly lower than 2% (Supplementary Fig. 4). The result is a non-uniform distribution of connectivity frequencies. We found that there are between nine and 12 overrepresented glomeruli (glomeruli with connectivity frequencies higher than 2%, *p*-value < 0.05) and between 10 and 13 underrepresented glomeruli (glomeruli with connectivity frequencies lower than 2%, *p*-value < 0.05) within each connectivity matrix. This result shows that the non-uniform distribution of connectivity frequencies is a feature present across species.

To compare the overall extent of bias in the distributions of connectivity frequencies derived from the connectivity matrices, we inferred their Jensen-Shannon distances. This statistical method measures the similarity of two probability distributions, with a distance of

zero indicating identical distributions. To gauge the extent to which the Jensen-Shannon distance indicates overall similarity of the observed distributions in connectivity frequencies, we compared the distributions of connectivity frequencies measured in the experimental matrices to those measured using the corresponding uniform shuffle matrices and obtained distances ranging from 0.23 to 0.27; when we compared the distributions of connectivity frequencies measured in the experimental matrices to those measured using the biased shuffle matrices, we obtained distances ranging from 0.08 to 0.09 (Supplementary Fig. 5). When we compared the distributions of connectivity frequencies measured using two *D. melanogaster* female matrices—one matrix was generated in this study and the other was generated in a previous study[29]—we obtained a relatively short distance of 0.17. Likewise, we obtained a relatively short distance of 0.15 when we compared the distributions measured in *D. melanogaster* females and males (Fig. 2c). This result suggests that the biases in connectivity are largely similar in both sexes. The distances measured when comparing *D. melanogaster* and *D. simulans* range from 0.16 to 0.17, indicating that the overall extent of bias in connectivity is similar in these species. By contrast, the distances between *D. melanogaster* and *D. sechellia* range from 0.20 to 0.22, whereas the distance between *D. simulans* and *D. sechellia* is 0.24, showing that the overall extent of biases in connectivity is higher in *D. sechellia* than in its sibling species. As *D. sechellia* and *D. simulans* are phylogenetically more closely related to each other than to *D. melanogaster*, the observed pattern suggests that the overall similarity in biases is not a function of evolutionary relatedness.

We next investigated whether the larger Jensen-Shannon distances measured for *D. sechellia* result from small differences in connectivity frequencies distributed across glomeruli or whether they result from large differences restricted to a few glomeruli. We first performed pairwise comparisons between *D. sechellia* and the generalists by using the ratio of connectivity frequencies obtained for a given glomerulus in the biased shuffle data sets (connectivity frequency measured in the *D. sechellia* matrix divided by the connectivity frequency measured in the biased shuffle data sets of all generalists) (Fig. 2d, e). We found that most glomeruli are connected to Kenyon cells at similar frequencies across all data sets but that a few glomeruli stand out when comparing *D. sechellia* to the generalists: the DL2d and DP1l glomeruli are significantly more connected in *D. sechellia* than in the generalists whereas the DC3 and VM5v glomeruli are significantly less connected in *D. sechellia* than in the generalists. To confirm these findings, we then performed pairwise comparisons and measured the ratio of connectivity frequencies obtained for a given glomerulus using the experimental data sets (connectivity frequency measured in matrix 1 divided by the connectivity frequency measured in matrix 2) (Fig. 2f). We could confirm that the DL2d, DP1l, DC3 and VM5v glomeruli show significant differences in their connectivity rates when comparing *D. sechellia* to one or both of the generalists.

The DL2d and DP1l glomeruli—which receive input from the acid-sensing Ir75b- and Ir75a-expressing neurons, respectively—were found to be connected at higher frequencies in *D. sechellia* than in one of the other two species (DL2d connectivity frequencies: *D. sechellia*: 3.19%, *D. simulans*: 0.68%, *p*-value < 0.01, *D. melanogaster*: 1.02%, *p*-value < 0.05; DP1l: *D. sechellia*: 3.19%, *D. melanogaster*: 0.44%, *p*-value < 0.001, *D. simulans*: 0.95%, *p*-value < 0.05). The DC3 and VM5v glomeruli were found to be connected at lower frequencies in *D. sechellia* than in one or both species. These glomeruli receive input from olfactory sensory neurons tuned to various fruit volatiles: the Or83c-expressing neurons associated with the DC3 glomerulus are narrowly tuned to farnesol, an odor made by yeast and citrus fruits, whereas the neurons associated with the VM5v (Or98a-expressing neurons) glomeruli are broadly tuned to alcohols and esters produced by fermenting fruits[13,14,31–33] (connectivity frequencies: DC3: *D. sechellia*: 1.37%, *D. melanogaster*: 4.50%, *p*-value < 0.001, *D. simulans*: 4.91%, *p*-value < 0.001; VM5v: *D.*

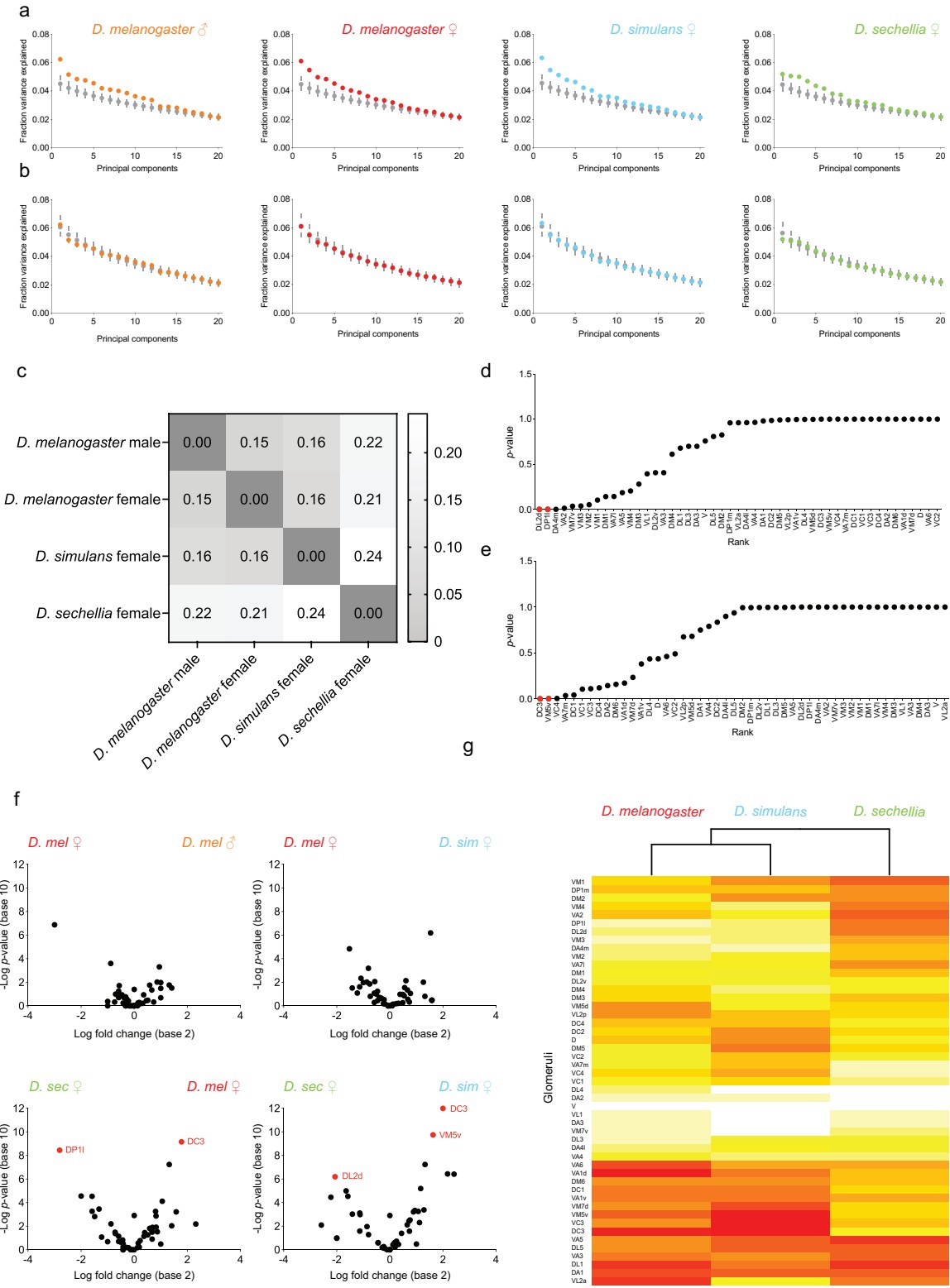

*sechellia*: 1.82%, *D. simulans*: 5.05%, *p*-value < 0.001, *D. melanogaster*: 3.36%, *p*-value < 0.05).

Overall, these results show that differences in connectivity frequencies are species-specific, not sexually dimorphic, and restricted to a fraction of olfactory channels. The channels that differ the most relay information about food odors to the mushroom body, suggesting that the representation ethologically relevant information in the mushroom body might change as species diverged. Most notably, the connectivity frequency measured for a given glomerulus changes in the *D. sechellia* lineage, being more similar in *D. melanogaster* and *D. simulans* than when comparing any of these species to *D. sechellia*. As *D. sechellia* is more closely related to *D. simulans* than to *D. melanogaster*, this pattern appears not to be a function of phylogenetic relatedness, but instead of ecological relatedness. To better visualize

**Fig. 2 | Shifts in connectivity biases across *Drosophila* species. a, b** Principal components were extracted using each connectivity matrix as well as 1000 uniform shuffle (**a**) or 1000 biased shuffle matrices (**b**); the fraction of the variance explained by each component was measured (*D. melanogaster* males: orange; *D. melanogaster* females: red; *D. simulans* females: blue; *D. sechellia* females: green; colors: experimental matrices, gray: shuffle matrices); error bars represent 95% confidence interval. **c** The Jensen-Shannon distances were measured by comparing the distributions in connectivity frequencies observed experimentally; the color bar denotes the length of the distances measured. See Supplementary Fig. 5 for the complete set of Jensen−Shannon distances. **d, e** For each glomerulus, the connectivity frequency measured in the *D. sechellia* matrix was compared to the average connectivity frequency obtained in a set of biased shuffle matrices generated using the generalist matrices, and the probability that a glomerulus being connected to Kenyon cells at a higher (**d**) or lower (**e**) frequency in the *D. sechellia* matrix was measured (*p*-value); glomeruli were ranked based on *p*-values and the

DL2d, DP1l, DC3 and VM5v projection neurons were further investigated (red). **f** The *p*-value measured for each glomerulus was plotted against the $\log_2$ fold change measured when comparing the connectivity frequencies measured for that glomerulus in the two matrices indicated on the plot. The statistical significance, or *p*-value, was measured for each glomerulus using the Fisher's exact test; to control for false positives, *p*-values were adjusted with a false discovery rate of 0.10 using a Benjamini-Hochberg procedure. Within these plots, a fold change with a value of 0 indicates that there is no shift in frequencies between matrices, whereas a fold change that is smaller or greater than 0 indicates that a given glomerulus is connected more frequently in one matrix than the other. Data points with *p*-values smaller than 0.01 are identified with a label (red); all other data points have *p*-values greater than 0.01 (black). **g** A clustering dendrogram based on the connectivity frequencies measured for each glomerulus across species. All source data used in this figure are provided in the Source Data file.

this notion, we generated a hierarchical clustering dendrogram based on the connectivity frequencies measured in the experimental matrices (Fig. 2g). In this dendrogram, each line represents a glomerulus, and the connectivity frequency measured in a given species for that glomerulus is depicted as a color gradation. Glomeruli were linked into clusters based on the similarity of their connectivity frequencies across species. In the tree constructed based on this analysis, *D. melanogaster* and *D. simulans* cluster and *D. sechellia* stands out as an outlier. This observation shows that changes in glomerular representation in the mushroom body are highly correlated with the ecology of a species, not its phylogeny, and, therefore, could have evolved in *D. sechellia* as this species diverged to exploit noni fruit.

## Two morphological features of projection neurons underlie shifts in biases

From our analyses of the above data sets, we observed two types of change in the mushroom body connectivity architecture of *D. sechellia* when compared to that of *D. melanogaster* and *D. simulans*: the representation of the DL2d and DP1l glomeruli increases and the representation of the DC3 and VM5v glomerulus decreases. In *D. melanogaster*, it is known that biases in connectivity are a function of the overall number of presynaptic boutons formed in the mushroom body by the projection neurons associated with a given glomerulus[25,26,28]. Thus, changes in glomerular representation across species could result from changes in the number of projection neurons associated with a given glomerulus, changes in the number of presynaptic boutons individual projection neurons form or both. To determine whether any of these cases prevail, we photo-labeled (Fig. 3a) and dye-labeled (Fig. 4a) the projection neurons innervating a given glomerulus to quantify the number of neurons and measure the volume of the presynaptic boutons the labeled neurons occupy in the mushroom body.

The DL2d and DP1l glomeruli are more frequently connected to Kenyon cells in *D. sechellia* (Fig. 2d, Supplementary Data 2). We identified eight DL2d projection neurons in *D. sechellia* (five neurons located in the anterior-dorsal cluster, or adDL2d neurons, and three neurons located in the ventral cluster, or vDL2d neurons) but only six DL2d projection neurons in *D. melanogaster* and *D. simulans* (five adDL2d neurons and one vDL2d neuron in both species) (Table 1, Fig. 3b, c). The number of DL2d projection neurons identified in *D. melanogaster* is consistent with the number of projection neurons reported in the available connectomes of the *D. melanogaster* brain[23] (Supplementary Fig. 6). In all species, the vDL2d neurons bypass the mushroom body and project only to the lateral horn. Therefore, the number of neurons connecting the DL2d glomerulus to the mushroom body is the same in all species. However, collectively, the adDL2d neurons show larger bouton volume in *D. sechellia* than they do in the other species (bouton volume of all adDL2d neurons: *D. melanogaster*: 112.18 ± 12.41; *D. simulans*: 104.44 ± 25.93; *D. sechellia*: 348.29 ± 186.63;

Table 1, Fig. 3d). We found that the increased number of connections between adDL2d neurons and Kenyon cells in *D. sechellia* is due to more presynaptic boutons being formed per neuron (Fig. 4b, c, Supplementary Table 1). We also found that individual adDL2d neurons show more complex branching patterns based on several quantifiable parameters (Fig. 4c, Supplementary Table 1). We identified a similar phenotype for the projection neurons associated with the DP1l glomerulus: there are three DP1l projection neurons in *D. sechellia* (one neuron located in the lateral cluster, or lDP1l neuron, and two vDP1l neurons) but only two DP1l projection neurons in *D. melanogaster* (one lDP1l neuron and one vDP1l neuron) as it has been reported in the connectomes[23] (Table 1, Supplementary Fig. 7). As with the vDL2d neurons, the vDP1l neurons bypass the mushroom body and only the lDP1l neurons connect to Kenyon cells. We found that for lDP1l neurons the bouton volume is larger in *D. sechellia* than in *D. melanogaster* and *D. simulans* (bouton volume of the lDP1l neuron: *D. melanogaster*: 244.39 ± 69.00; *D. simulans*: 232.41 ± 100.73; *D. sechellia*: 388.91 ± 75.88).

The DC3 and VM5v glomeruli are less frequently connected to Kenyon cells in *D. sechellia* (Fig. 2d, Supplementary Data 2). We identified two DC3 projection neurons in *D. sechellia* (two adDC3 neurons) but as many as three in *D. melanogaster* (three adDC3 neurons as reported in the connectomes[23]) and six in *D. simulans* (three adDC3 neurons and three vDC3 neurons) (Table 1, Fig. 3b, c, Supplementary Fig. 8). In *D. simulans*, the vDC3 neurons bypass the mushroom body and project only to the lateral horn. The volume of presynaptic boutons collectively formed by the adDC3 neurons in the mushroom body is significantly smaller in *D. sechellia* than in *D. melanogaster* and *D. simulans* (bouton volume of all adDC3 neurons: *D. melanogaster*: 370.15 ± 195.68; *D. simulans*: 364.06 ± 146.60; *D. sechellia*: 233.02 ± 83.80; Table 1, Fig. 3d). However, individual adDC3 projection neurons are morphologically similar across species with no significant differences in branch length, number of forks or bouton number (bouton volume of individual adDC3 neuron: *D. melanogaster*: 184.15 ± 34.60; *D. simulans*: 169.01 ± 43.23; *D. sechellia*: 165.36 ± 25.56; Fig. 4b, c, Supplementary Table 1). Thus, the decrease in the number of connections between adDC3 neurons and Kenyon cells in *D. sechellia* is due to a decrease in number of projection neurons innervating the DC3 glomerulus in that species. We identified a similar phenotype for the projection neurons associated with the VM5v glomerulus: there are two adVM5v projection neurons in *D. sechellia* but as many as three adVM5v projection neurons in *D. melanogaster*, as reported in the connectomes[23], and three adVM5v projection neurons in *D. simulans* (Table 1, Fig. 3b, c, Supplementary Fig. 6). We found that, as for the adDC3 neurons, the adVM5v neurons form collectively a smaller bouton volume in *D. sechellia* but, individually, the VM5v neurons are similar across species (bouton volume of all VM5v neurons: *D. melanogaster*: 256.78 ± 104.09; *D. simulans*: 268.43 ± 132.67; *D. sechellia*: 140.14 ± 77.76; Table 1; bouton volume of individual VM5v

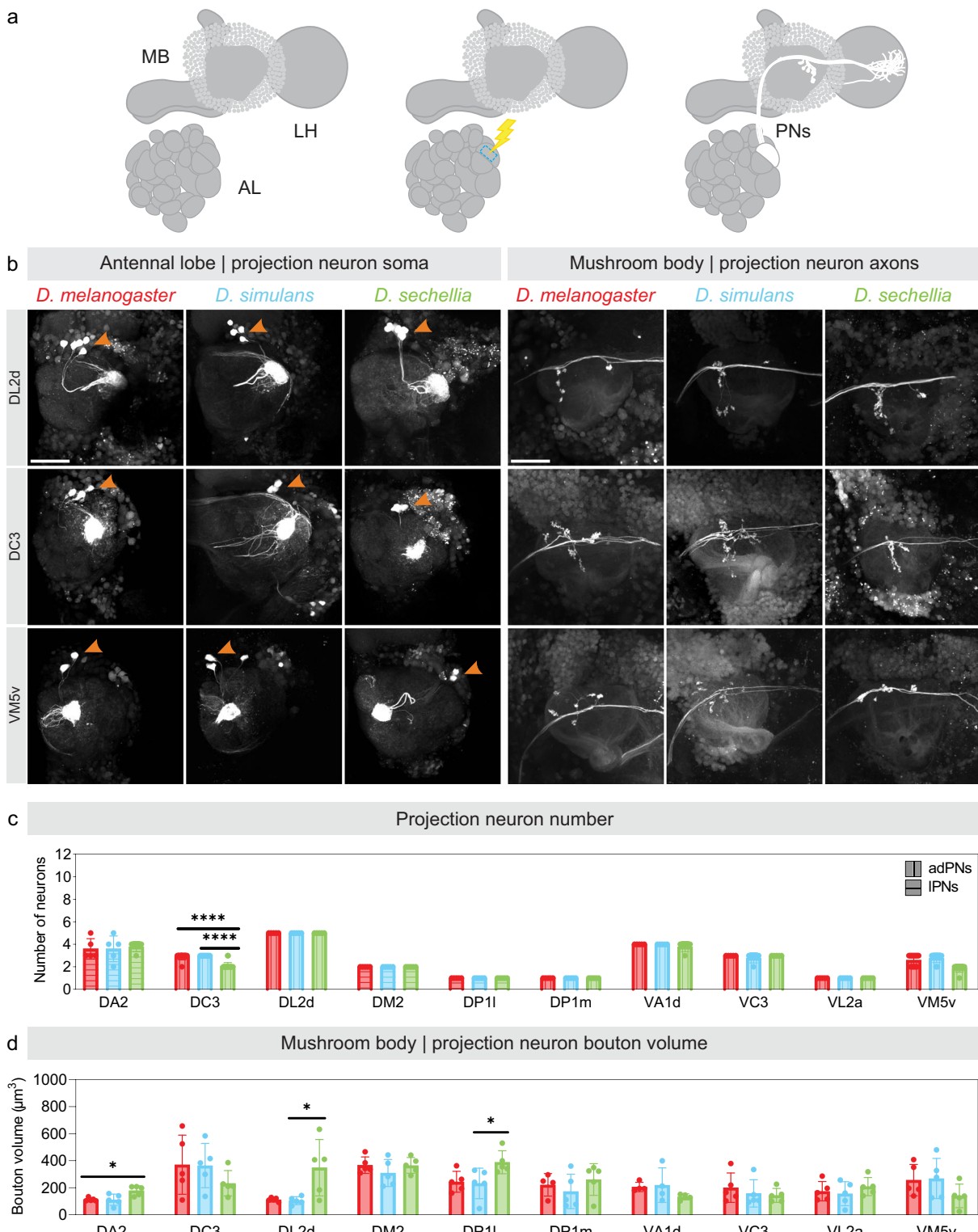

**Fig. 3 | Morphological features of the projection neurons innervating a given glomerulus in different *Drosophila* species. a** Schematic depicting the technique used to photo-label the projection neurons innervating a given glomerulus: a glomerulus is used as a landmark for photo-labeling (blue dashed outline), and the projection neurons connected to the targeted glomerulus are photo-labeled after successive rounds of photo-labeling. **b** The projection neurons innervating the DL2d (upper panels), DC3 (middle panels) and VM5v (lower panels) glomeruli were photo-labeled in *D. melanogaster* (left column of each panel), *D. simulans* (middle column of each panel) and *D. sechellia* (right column of each panel); the cell bodies of these neurons (left panels) and the axonal termini that these neurons extend in the mushroom body (right panels) were imaged. Scale bar is 50 μm. **c**, **d** The number of photo-labeled neurons (c) and the volume of the presynaptic boutons these neurons form in the mushroom body (d) were quantified and compared across species (red: *D. melanogaster*; blue: *D. simulans*; green: *D. sechellia*). The statistical significance, or *p*-value, was measured using the Mann–Whitney *U* test (*$p$-value < 0.5, **$p$-value < 0.01, ***$p$-value < 0.001; $n = 5$, standard deviation from mean is shown). See Table 1 for quantifications. All source data used in this figure are provided in the Source Data file.

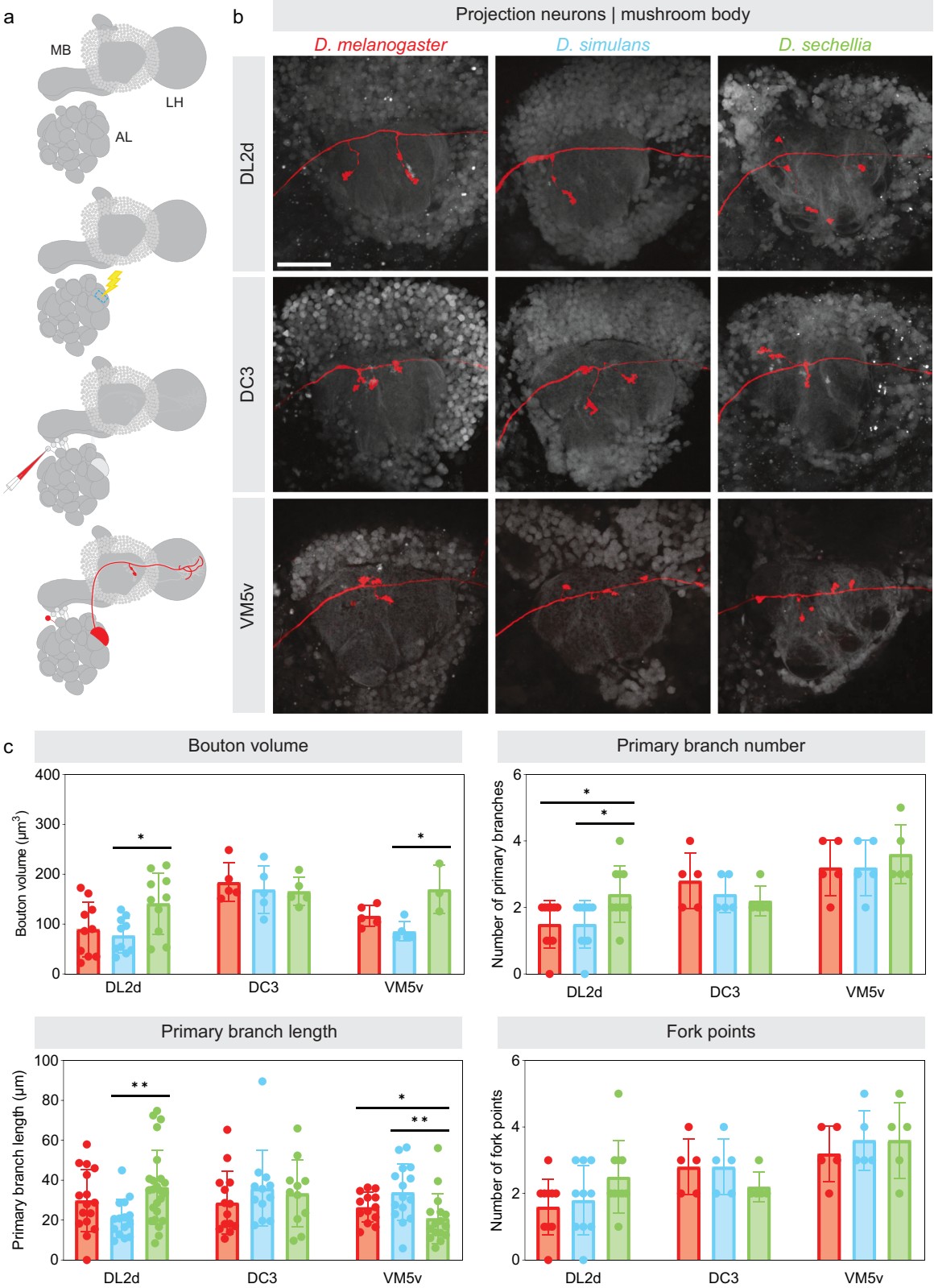

neurons: *D. melanogaster*: 116.45 ± 18.68; *D. simulans*: 85.38 ± 17.48; *D. sechellia*: 169.73 ± 39.67; Fig. 4b, c, Supplementary Table 1). Thus, as with the adDC3 neurons, there are fewer VM5v neurons in *D. sechellia*, and, therefore, fewer connections between the VM5v glomerulus and Kenyon cells. Combined with the observations made for the DL2d and DP1l projection neurons, these results suggest that increases in glomerular representation in *D. sechellia* occur through increases in

bouton number, whereas decreases in glomerular representation occur through decreases in the number of projection neurons associated with different glomeruli. These observations further suggest that selection pressures may influence shifts in glomerular representation through at least two different types of molecular mechanism – those regulating synaptogenesis and those regulating neurogenesis – and that these mechanisms could be glomerular specific.

**Fig. 4 | Morphological features of individual projection neurons in different *Drosophila* species. a** Schematic depicting the technique used to dye-label a projection neuron innervating a given glomerulus: a glomerulus is used as a landmark for a first round of photo-labeling (blue dashed outline) during which the projection neurons connected to the targeted glomerulus are lightly photo-labeled; dye is electroporated in one of the photo-labeled projection neurons such that a single projection neuron is dye-labeled. **b** A projection neuron innervating the DL2d (upper row), DC3 (middle row) and VM5v (lower row) glomeruli were dye-labeled in *D. melanogaster* (left column), *D. simulans* (middle column) and *D. sechellia* (right column); the axonal termini these neurons extend in the mushroom body were imaged. Scale bar is 50 μm. **c** Various morphological features displayed by projection neurons in the mushroom body were quantified and compared across species (red: *D. melanogaster*; blue: *D. simulans*; green: *D. sechellia*). The statistical significance, or *p*-value, was measured using the Mann–Whitney *U* test (*$p$-value < 0.5, **$p$-value < 0.01, ***$p$-value < 0.001; $n = 5$, standard deviation from mea*n* is shown). See Supplementary Table 1 for quantifications. All source data used in this figure are provided in the Source Data file.

**Table 1 | Morphological features of photo-labeled projection neurons across species**

| | Neuron number | | | Bouton cluster volume (μm3) | | |
|---|---|---|---|---|---|---|
| | *D. melanogaster* | *D. simulans* | *D. sechellia* | *D. melanogaster* | *D. simulans* | *D. sechellia* |
| **DA2** | 4 ± 0.80 lPNs<br>0 vPN | 4 ± 1.02 lPNs<br>0 vPN | 4 ± 0.40 lPNs<br>0 vPN | 112.79 ± 11.30 | 110.95 ± 37.06 | 179.18 ± 31.76 |
| **DC3** | 3 ± 0.29 adPNs<br>0 vPN | 3 adPNs<br>3 vPNs | 2 ± 0.29 adPNs<br>0 vPN | 370.15 ± 195.68 | 364.06 ± 146.60 | 233.02 ± 83.80 |
| **DL2d** | 5 adPNs<br>1 vPN | 5 adPNs<br>1 vPN | 5 adPNs<br>3 vPNs | 112.18 ± 12.41 | 104.44 ± 25.93 | 348.29 ± 186.63 |
| **DM2** | 2 lPNs<br>0 vPN | 2 lPNs<br>0 vPN | 2 lPNs<br>0 vPN | 367.38 ± 54.87 | 309.52 ± 89.42 | 365.40 ± 52.39 |
| **DP1l** | 1 lPN<br>1 vPN | 1 lPN<br>2 vPNs | 1 lPN<br>2 vPNs | 244.39 ± 69.00 | 232.41 ± 100.73 | 388.91 ± 75.88 |
| **DP1m** | 1 adPN<br>0 vPNs | 1 adPN<br>0 vPNs | 1 adPN<br>0 vPNs | 221.59 ± 72.28 | 172.85 ± 113.80 | 260.95 ± 105.19 |
| **VA1d** | 3 adPNs<br>1 vPN | 3 adPNs<br>1 vPN | 3 ± 0.40 adPNs<br>0 vPN | 205.89 ± 26.65 | 195.78 ± 109.79 | 155.87 ± 50.25 |
| **VC3** | 3 adPNs<br>0 vPN | 3 adPNs<br>0 vPN | 3 adPNs<br>0 vPN | 200.70 ± 97.03 | 158.48 ± 90.40 | 143.59 ± 46.17 |
| **VL2a** | 1 adPN<br>3 vPNs | 1 adPN<br>3 vPNs | 1 adPN<br>3 vPNs | 174.36 ± 64.22 | 156.05 ± 79.93 | 208.92 ± 60.55 |
| **VM5v** | 3 ± 0.49 adPNs<br>0 vPN | 3 ± 0.40 adPNs<br>0 vPN | 2 ± 0.33 adPNs<br>0 vPN | 256.78 ± 104.09 | 268.43 ± 132.67 | 140.14 ± 77.76 |

Morphological features of projection neurons—namely the number of neurons associated with a given glomerulus and the volume of the presynaptic sites or boutons these neurons form in the mushroom body—were measured and compared across species ($n = 5$ for each type of projection neuron, standard deviation from mean is shown). Projection neurons showing significant shifts in connectivity frequencies (DC3, DL2d, VM5v, DP1l, VA1d, VC3, and VL2a projection neurons) were analyzed as well as some projection neurons that did not show significant shifts in connectivity frequencies (DA2, DM2, and DP1m projection neurons). Projection neurons were typed based on whether their cell bodies are located in the anterior-dorsal (ad), lateral (l) or ventral cluster (v). All source data used in this table are provided in the Source Data file.

We extended our anatomical analyses beyond the DL2d, DP1l, DC3 and VM5v projection neurons. We did not detect significant inter-specific differences for the VA1d, VC3 and VL2a projection neurons although we identified these glomeruli as differently represented in at least one of the species investigated (Table 1, Supplementary Fig. 7). We have also performed anatomical analyses on neurons innervating glomeruli that we identified as identically represented in all three species, namely the DA2, DM2 and DP1m glomeruli. We could not detect any noticeable differences in the axonal termini that these projection neurons extend in the mushroom body (Table 1, Supplementary Fig. 7). Although our photo-labeling and dye-labeling techniques might not fully capture the extent of morphological differences that exist between the species investigated, our results suggest that our approach is sufficient to identify the most significant morphological features that underlie shifts in connectivity across species. Altogether, our results suggest that interspecific differences in the mushroom body connectivity architecture are specific and restricted to a small number of projection neurons.

**Connectivity frequencies and learning performance**
We tested whether the changes in glomerular representation we observed in the mushroom body of the species correlates with differences in learning performance by using a well-established behavioral paradigm[34]. In this paradigm, flies were conditioned to associate a stimulus—either air perfumed with an odor diluted in mineral oil or pure mineral oil—with electric shocks, and their preference for the conditioned stimulus was subsequently tested in a T-maze (Supplementary Fig. 8). In short, we measured the number of flies seeking out the conditioned stimulus over the number of flies seeking out the unconditioned stimulus to derive a Performance Index. We used either a protocol that included a single regimen of electric shocks (single training) or a protocol that included six spaced regimens of electric shocks (spaced training). In a first series of experiments, we tested odors known to activate the glomeruli whose representation shifts most drastically across species: the DC3 and DL2d glomeruli. Farnesol, strongly and selectively activates the OR83c-expressing neurons in *D. melanogaster*[32] (Supplementary Table 2). The OR83c-expressing neurons project to the DC3 glomerulus, and the adDC3 projection neurons connect to Kenyon cells at a much higher frequency in *D. melanogaster* and *D. simulans* than in *D. sechellia* (Fig. 2e, f, Supplementary Table 2). We found that farnesol could not trigger strong learned responses in any of the species in either the single or spaced training protocol (Fig. 5a, b, Supplementary Fig. 9). Hexanoic acid, a noni odor, weakly activates multiple types of olfactory sensory neuron in *D. melanogaster* but strongly activates the IR75b-expressing neurons in *D. sechellia*[21,33] (Supplementary Table 2). The IR75b-expressing neurons project to the DL2d glomerulus, and the adDL2d projection neurons connect with Kenyon cells at a high frequency in *D. sechellia*; the projection neurons associated with the glomeruli activated by hexanoic acid collectively connect at a similarly high frequency in *D. melanogaster* and *D. simulans* (Supplementary Table 2). We found that hexanoic acid can trigger learned responses in *D. melanogaster* and *D. simulans* when using

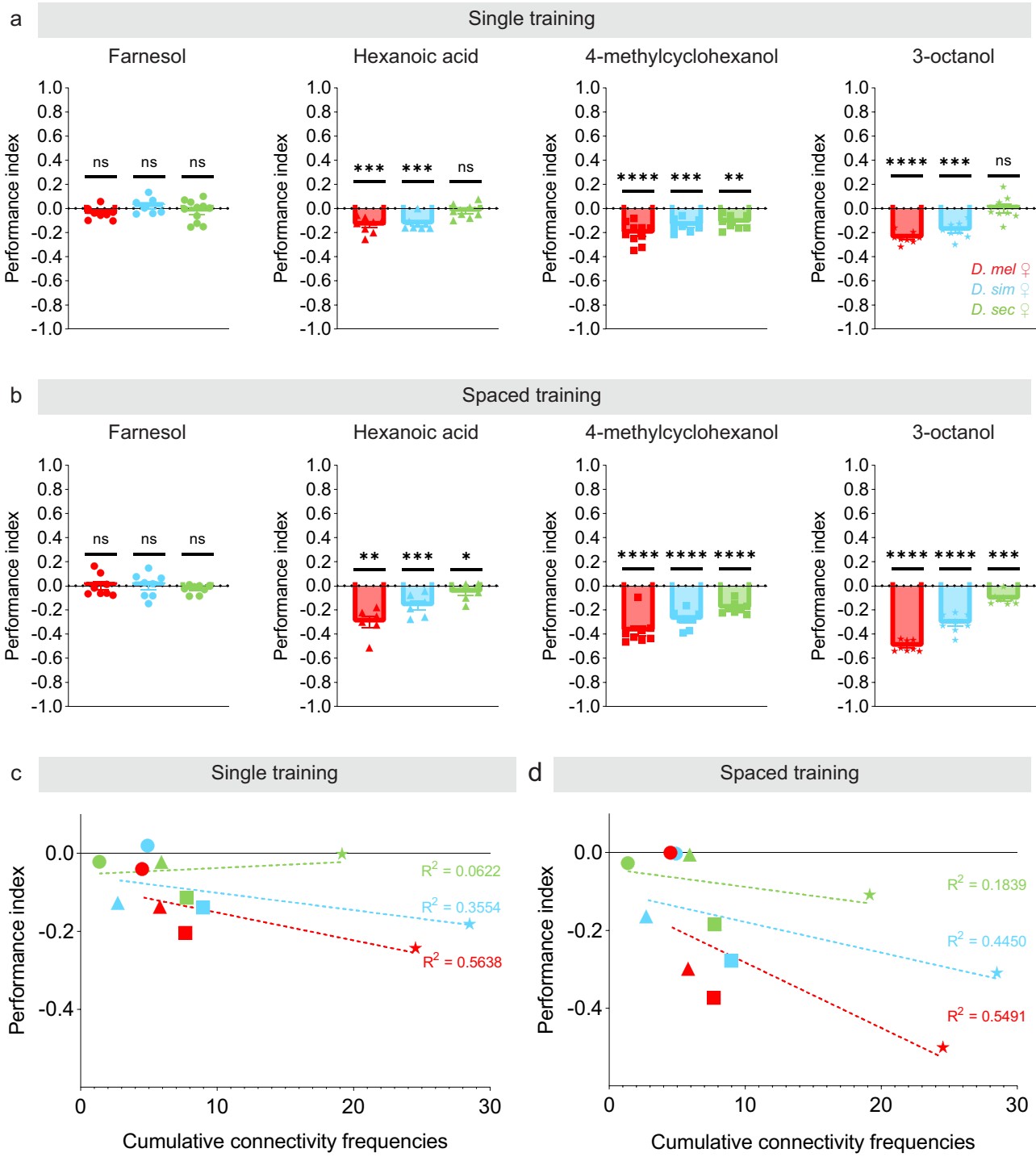

**Fig. 5 | Learning abilities differ across species. a, b** Flies (*D. melanogaster*: red; *D. simulans*: blue; *D. sechellia*: green) were trained to associate an odor (farnesol: circles, hexanoic acid: triangles, 4-methylcyclohexanol: squares or 3-octanol: stars) or its solvent (mineral oil) with punitive electric shocks using a single regimen of shocks (**a**) or six regimens of shocks (**b**) and learning was measured as a Performance Index; the Performance Indices obtained for the odor-pairing and the reciprocal pairing was averaged. See Supplementary Fig. 9 for individual Performance Indices. The statistical significance, or *p*-value, was measured using the sample *t* test using 0 as the hypothetical mean (\**p*-value < 0.5, \*\**p*-value < 0.01, \*\*\**p*-value < 0.001, \*\*\*\**p*-value < 0.0001; $n \geq 7$, standard deviation from mean is shown). (**c**, **d**) The Performance Indices obtained for a given odor in a particular species (farnesol: circles, hexanoic acid: triangles, 4-methylcyclohexanol: squares or 3-octanol: stars) were plotted against the cumulative frequencies of the glomeruli known to be activated by a particular odor (based on a previous study[33]; Supplementary Table 2); the $R^2$ values obtained for each regression line are shown. All source data used in this figure are provided in the Source Data file.

either the single or spaced training protocol but not in *D. sechellia* where weak learned responses of much smaller amplitudes are only observed when using the spaced training protocol (Fig. 5a, b, Supplementary Fig. 9). These results suggest that activation of a single glomerulus is insufficient to elicit robust learned responses in this species, and that the activation of multiple glomeruli that collectively connect to Kenyon cells at high frequencies might be required for learning.

We thus set out to test odors known to activate multiple types of olfactory sensory neuron in *D. melanogaster*, and therefore multiple

glomeruli: 4-methylcyclohexanol and 3-octanol, which are detected by at least three and nine different types of olfactory sensory neuron, respectively[31,33] (Supplementary Table 2). We found that both odors triggered strong learned responses in *D. melanogaster* using either the single or spaced training protocol (Fig. 5a, b, Supplementary Fig. 9). We noticed that the magnitude of the Performance Index varies significantly across odors: 3-octanol, which activates the largest number of glomeruli, gave rise to larger Performance Indices than 4-methycyclohexanol. To determine whether differences in connectivity frequencies could underlie the amplitude of the Performance Indices, we plotted the cumulative connectivity frequencies of the glomeruli known to be activated by each odor in *D. melanogaster* against the Performance Indices measured with a given protocol (Fig. 5c, d). We observed a strong positive correlation, suggesting that odors activating a large number of highly connected glomeruli are more learnable than odors activating one or a few glomeruli. We performed similar analyses on *D. simulans* and *D. sechellia* and observed similar correlations. It is important to note that the cumulative connectivity frequencies were calculated using the odor responses measured in *D. melanogaster*, with the exception of hexanoic acid for which recordings collected in *D. simulans* and *D. sechellia* are available[21]. Interestingly, the Performance Indices also vary across species although all species display equally strong avoidance to electric shocks (Supplementary Fig. 10): *D. melanogaster* performed better than *D. simulans* and remarkably better than *D. sechellia* regardless of the protocol used, whereas *D. sechellia* was only able to learn when we used the spaced training protocol (Fig. 5c, d). These differences could reflect the fact that this training paradigm was originally developed and optimized for *D. melanogaster*; our data set indeed represents one of the first comparative study investigating differences in learning performance across *Drosophila* species. These differences could also reflect the fact that *D. sechellia* is impaired in its ability to synthesize dopamine, a neurotransmitter essential for learning[35]. The poor ability of *D. sechellia* to learn may reveal a relaxed requirement for this species to form associations with a wide range of stimuli as, unlike generalists, its survival depends on a single resource, noni. To test whether *D. sechellia* can learn any odor, we trained flies with an odor predicted to activate a large number of glomeruli that are connected to Kenyon cells at high frequencies in *D. sechellia*, namely 2,3-butanedione and isopentyl acetate using the single training protocol (Supplementary Fig. 11, Supplementary Table 2). These odors elicited robust learning in all species suggesting that *D. sechellia* are capable of learning when a large number of projection neurons are recruited. Altogether, these results suggest that increased connectivity between projection neurons and Kenyon cells enhances learning performance.

## Discussion

In this study, we harness a phylogenetically and ecologically informed approach to pinpoint differences in neuronal connectivity architecture and learning performance between three species of *Drosophila*. The species differ in the frequency with which inputs from a small set of olfactory channels—mainly those relaying information about food odors—are represented among the overall input to a higher-order processing center, the mushroom body. Notably, most of these differences are found in the specialist species *D. sechellia*, suggesting they are due to an ecological niche shift, rather than merely a function of phylogenetic distance. Evolutionary differences in sensory representation are caused by morphological alterations of the projection neurons connecting individual glomeruli to the mushroom body, either through changes in the number of projection neurons or through changes in the number of presynaptic boutons per neuron. Our study also suggests that increased connectivity might enhance learning performance in an associative task.

While we have identified clear species-specific connectivity patterns, it is likely that other, more nuanced features of connectivity architecture underlie the evolution of the mushroom body in *Drosophila* species. A recent electron microscopy-based study of the *D. melanogaster* mushroom body has revealed a subtle but significant connectivity pattern between a subgroup of projection neurons and the α/β and α′/β′ Kenyon cells, but our mapping technique is not sensitive enough to capture this particular feature of connectivity architecture[28,29]. Likewise, our study has limitations regarding the conclusions that can be drawn about the forces driving the evolution of connectivity architecture in the mushroom body. For instance, it remains unclear whether the neuronal connectivity traits identified are variable across individuals in natural populations, especially those inhabiting different environments. It remains equally unclear whether the anatomical and behavioral changes that we observed confer a fitness advantage and could be adaptive. The fact that we were able to correlate a visible neuronal phenotype with these connectivity changes and correlate connectivity frequencies with learning performance, makes such investigations theoretically possible in the future. Another important limitation of our study is the assumption that olfactory sensory neurons have the same detection capabilities across all species. However, a recent study indicates that the activity patterns evoked by odors in the antennal lobe of the species we investigated, while not identical, are largely conserved[17].

Despite these limitations, our study contributes new insights into the emergence of behavioral adaptations. Behavioral adaptations arise from modifications in the way neuronal circuits process information, and it is known that different cellular mechanisms can give rise to such modifications[5,36,37]. For instance, changes in the expression levels of receptors or ion channels can alter the electrophysiological properties of neurons and, consequently, neuronal output. There are many documented examples of such evolutionary changes at the periphery, including in the *Drosophila* olfactory systems, showing that sensory systems can adapt by finely tuning their detection capabilities to features of the environment peculiar to a species[5,36]. Changes in neuronal connectivity, whether through changes in synaptic weights between existing partners or through the formation of new synaptic partners, can also alter the way information flows in a circuit. In principle, such modifications can occur at any level within a circuit. However—perhaps because neurons at the periphery are more accessible than neurons embedded in the higher brain centers—there is a predominance of documented functional changes in sensory neurons over connectivity changes in central nervous systems. Our study provides the first evidence at the cellular level for such evolutionary changes in the connectivity architecture of higher olfactory brain centers and how these changes might enhance cognitive functions.

While it is understood that behaviors and their underlying neuronal circuits evolve through various mechanisms, our limited examples of these mechanisms hinder our capacity to determine if some are more common than others. Namely, how higher brain centers evolve remains, for the most part, completely unknown. Higher brain centers often feature more integrated circuits with many connections, and therefore it is conceivable that they might be less liable to change during evolution due to the disruptive impacts such changes might have across circuits. The anatomical changes we observed in this study are comparatively subtle, involving changes in the quantity of neuronal connections formed while leaving the overall random circuit architecture intact. Theoretical studies have demonstrated that randomization of input enables mushroom body-like networks to generate a representation space of high dimensionality, where many odors can be represented by non-overlapping neuronal ensembles[38–40]. We propose that biases in connectivity enable the mushroom body to better represent ecologically meaningful odors by increasing the coding space allocated to these odors, potentially increasing the fitness of an animal. Shifts in connectivity biases emphasize information streams that are most relevant to an animal without completely inactivating or

adding de novo streams. This evolutionary pattern may be widespread across brain centers and species.

## Methods

### Fly stocks and husbandry

Flies were reared under standard conditions (25 °C, 60% humidity) in incubators that maintain a 12 h light/12 h dark cycle (Percival Scientific Inc, Cat#DR36VL); *D. melanogaster* and *D. simulans* flies were reared on standard cornmeal agar medium, whereas *D. sechellia* flies were reared on standard cornmeal agar medium that was supplemented with noni juice (Healing Noni). The stocks used and their sources were as follows: *D. melanogaster*: *w*[1118];;; (Bloomington Stock Center, 5905), *yw*;[N-Synaptobrevin-GAL4][2-1];; (J. Simpson, University of California, Santa Barbara)[24] and *y*[1],*w*[1118];[10xUAS-IVS-Syn21-mC3PA-GFP-p10][attP40];; (Axel laboratory, Columbia University)[24]; *D. sechellia*: *w*[N-Synaptobrevin-GAL4, w*]3P3-RFP-DEL;;; and *w*[UAS-C3PA-GFP, w*]3P3-RFP-DELA;;; (Benton laboratory, University of Lausanne)[16]; *D. simulans*: *attp*[2176] and *attp*[2178] (Stern laboratory, Janelia Farm Research Campus)[41].

### Transgenesis

A *N-synaptobrevin-GAL4* (*D. simulans*) plasmid was generated by cloning a 1.9 kb sequence upstream of *N-synaptobrevin* using PCR and genomic DNA extracted from *D*. simulans *attp*[2176] flies (forward oligonucleotide: GATCGGTACCGAACTCGTCCTCAAAGATGGAAACAGAG; reverse oligonucleotide: GATCGCGGCCGCGAATTCGGCTGGCGATGATTAGGATG); the amplified sequence was inserted into the pGal4attB plasmid using the NotI and KpnI restriction sites; this *N-synaptobrevin-GAL4* (*D. simulans*) plasmid was injected into the *D.* simulans *attp*[2176] strain with the φC31 integrase following a standard protocol (BestGene) resulting in the *y*[1]*w*[1]; *pBac{3XP3::EYFP, N-synaptobrevin-GAL4}*[attp2176];; transgenic line[42]. The *y*[1]*w*[1];;*pBac{3XP3::EYFP, UAS-C3PA-GFP}*[attp2178] transgenic line was generated by injecting a *UAS-C3PA-GFP* plasmid into the *D. simulans attp*[2178] strain using similar protocols[43].

### Reconstructing antennal lobes

Antennal lobes were reconstructed from confocal images of immuno-stained brains. The brains of flies were dissected at room temperature in a phosphate buffered saline solution or PBS (Sigma-Aldrich, P5493), fixed in 2% paraformaldehyde (Electron Microscopy Sciences, 15710) for either 45 min (*D. melanogaster*) or 35 min (*D. simulans* and *D. sechellia*) at room temperature, washed five times in PBST (PBS with 1x Triton, Sigma-Aldrich, T8787) at room temperature, blocked with 5% goat Serum (Jackson ImmunoResearch Laboratories) in PBST for 30 min at room temperature, and incubated in a solution that contained the primary antibody (1:20 in 5% Goat Serum/PBST, Developmental Studies Hybridoma Bank, nc82, AB 2314866) at 4 °C overnight. On the following day, brains were washed four times in PBST and incubated in a solution that contained the secondary antibody (1:500 in 5% Goat Serum/PBST, Thermal Fisher, goat anti-mouse Alexa Fluor 488, AB 2576217) at 4 °C overnight. On the following day, brains were washed four times in PBST and mounted on a slide (Fisher Scientific, 12-550-143) using the mounting media VECTASHIELD (Vector Laboratories Inc., H-1000). Immuno-stained brains were imaged using an LSM 880 confocal microscope (Zeiss). Each antennal lobe was reconstructed from a confocal image using the segmentation software Amira (FEI Visualization Sciences Group, version 2020.3.1). Individual glomeruli were reconstructed via manual segmentation: boundaries were demarcated by hand and interpolated. Glomeruli were assigned identities according to their position based on the available anatomical maps and the *D. melanogaster* hemibrain connectome v1.2.1[13,14,30,44]. Glomerular volumes were calculated from the reconstructed voxel size, and the sum of those volumes were used to calculate whole antennal lobe volumes. We identified a total of 51 glomeruli in the antennal lobe reconstructions but only 49 in the mapping experiments

used to generate the connectivity matrices. This is because VC3 is split into two glomeruli − VC3m and VC3l − in the reconstructions but when scoring matrices, we could not distinguish VC3m from VC3l.

### Photo-labeling projection neurons and Kenyon cells

Neurons were photo-labeled based on a previously published protocol[45]. In short, brains were dissected in saline (108 mM NaCl, 5 mM KCl, 5 mM HEPES, 5 mM Trehalose, 10 mM Sucrose, 1 mM $NaH_2PO_4$, 4 mM $NaHCO_3$, 2 mM $CaCl_2$, 4 mM $MgCl_2$, pH≈7.3), treated for 1 min with 2 mg/ml collagenase (Sigma-Aldrich) and mounted on a piece of SYLGARD placed at the bottom of a Petri dish. Each brain was either mounted with its anterior side facing upward (for photo-labeling projection neurons) or with its posterior side facing upward (for photo-labeling Kenyon cells). The photo-labeling and image acquisition steps were performed using a two-photon laser scanning microscope (Bruker, Ultima) with an ultrafast Chameleon Ti:Sapphire laser (Coherent) modulated by Pockels Cells (Conotopics). During the photo-labeling step, the laser was tuned to 710 nm and about 5 to 30 mW of laser power was used; during the image acquisition step, the laser was tuned to 925 nm and about 1 to 14 mW of laser power was used. Both power values were measured behind the objective lens. A 60X water-immersion objective lens (Olympus) was used for both photo-labeling and image acquisition. A GaAsP detector (Hamamatsu Photonics) was used for measuring green fluorescence. Photo-labeling was performed by drawing a region of interest−on average 1.0 ×1.0 μm −either in the center of the targeted glomerulus (for labeling projection neurons) or in the center of the soma (for labeling Kenyon cells). Photo-labeling projection neurons: Photoactivation was achieved through two to four cycles of exposure to 710-nm laser light, during which each pixel was scanned four times, with 25 repetitions per cycle, and 15 min rest period between each cycle. Image acquisition was performed with the laser tuned to 925 nm at a resolution of 512 by 512 pixels with a pixel size of 0.39 μm and a pixel dwell time of 4 μs; each pixel was scanned twice. A minimum of five samples per species were analyzed for each type of projection neuron. Photo-labeling Kenyon cells: Photoactivation was achieved through three to five single scans with the laser tuned to 710 nm, during which each pixel was scanned eight times. Before image acquisition, a 10 min rest period was implemented to allow diffusion of the photoactivated fluorophore within the neuron. Image acquisition was performed at a resolution of 512 by 512 pixels with a pixel size of 0.39 μm and a pixel dwell time of 4 μs; each pixel was scanned 2 times.

### Mapping Kenyon cell input connections using dye electroporation

The projection neurons connecting to a photo-labeled Kenyon cell were identified based on previously published protocols[26,29]. See Supplementary Fig. 3 for a schematic depicting the procedure. In short, electrodes were made by pulling borosilicate glass pipette with filament (Sutter Instruments, BF100-50-10) to a resistance of 9-11 MΩ, fire-polished using a micro-forge (Narishige) to narrow their opening and backfilled with 100 mg/ml 3000-Da Texas-dextran dye (Thermo-Fisher, D3328). Under the guidance of a two-photon microscope (Bruker, Ultima), an electrode was centered into the post-synaptic terminal−or claw−of a photo-labeled Kenyon cell using a motorized micromanipulator (Sutter Instruments). Short current pulses (each 10-50 V in amplitude and 0.5 millisecond long) were applied until the projection neuron connecting to the targeted Kenyon cell claw was visible. Not all the projection neurons connecting to a given Kenyon cells were dye-filled but on average 4 ± 1 of the claws formed by a given Kenyon cell were dye-filled. An image of the antennal lobe was acquired at the end of the procedure. Dye-labeled glomeruli were identified based on their shape, position and the location of their soma as defined in the available anatomical maps and the *Drosophila melanogaster* hemibrain connectome v1.2.1[13,14,30,44].

The selection of *D. melanogaster* male connectivity matrices for this study was driven by specific methodological considerations. The transgenes used for the mapping technique, *nSynaptobrevin-GAL4* and *UAS-photoactivatable-GFP*, are both located on the X chromosome in *D. sechellia*, limiting the analysis to female flies. Initially, the reference matrix was the male *D. melanogaster* connectivity matrix from Caron *et al.* (2013) Nature. To ensure consistency and comparability, a matrix for male *D. melanogaster* was first generated. To assess any potential sexual dimorphism in connectivity biases, a female *D. melanogaster* was generated and compared to the male matrices. No meaningful differences were found and therefore species-comparisons were performed using female matrices only.

### Dye-labeling individual projection neurons

Individual projection neurons were dye-labeled using a previously published protocol[16,43]. The cell body of the projection neuron of interest was first identified by lightly photo-labeling all the projection neurons innervating a given glomerulus by performing a single cycle of exposure to 710 nm light. After a rest period of 10 min, an unpolished electrode filled with Texas Red dextran dye was attached to the cell body of one of the photo-labeled projection neurons, and the dye was electroporated into the neuron using short current pulses; each pulse was 10 to 30 V in amplitude and 0.5 millisecond long. A resting period of about 30 min allowed the dye to diffuse throughout the neuron. Image acquisition was performed at a resolution of 512 by 512 pixels with a pixel size of 0.39 μm and a pixel dwell time of 4 μs; each pixel was scanned two times. A minimum of five samples per species were analyzed for each type of projection neuron.

### Quantifying morphological features of projection neurons

Representative images of projection neurons were projected at maximal intensity using the ImageJ/Fiji software[46] (National Institutes of Health). Projection neurons were counted based on the number of photo-labeled cell bodies observed in the anterior or lateral or ventral clusters of the antennal lobe. Primary branches were defined as processes that emerge from the main axonal projection that traverses the calyx of the mushroom body. The length of the branches formed by a projection neuron and the number of forks were quantified using the Simple Neurite Tracer plugin for ImageJ/Fiji software, while the surface area of the axonal arbors in the mushroom body calyx and lateral horn was calculated using the ROI Manager and Measure features of this software[47]. Total projection neuron bouton volume for a given sample was measured using Fluorender (University of Utah Scientific Computing and Imaging Institute; version 2.26.2[47,48]): boutons were traced using the Paint Brush function. To distinguish boutons from the background, the Edge Detect parameter was kept on and the Edge STR was fixed at 0.505, while the selection threshold was adjusted to different values depending on signal intensity[48]. The Physical Size value of the traced boutons was reported as total bouton volume.

### Hierarchical clustering dendrogram

The hierarchical clustering dendrogram was generated using RStudio and the gplots package. The connectivity frequencies measured for each glomerulus in a given species (based on the female connectivity matrices) were used as data. The Euclidean distance between connectivity frequencies was measured as a proxy for similarity. A tree representing overall similarity was generated by clustering glomeruli using the Complete Linkage function.

### Aversive learning paradigm

The aversive learning paradigm was designed based on a previously published protocol[49]. See Supplementary Fig. 8 for a schematic depicting the procedure. Flies were collected a few hours before performing the protocol and housed in regular food vials before being tested. Groups of flies—containing between 60 and 100 individuals—were introduced in a T-maze training apparatus (CelExplorer Labs Co., TMK-501) that was attached to a flowmeter that kept a constant stream of 0.7 L/min (Dwyer Instruments, 116011-01). During the training phase, flies were exposed to a first stimulus, henceforth referred to as the Conditioned Stimulus + (CS + ), at the same time as they were subjected to a regimen of electric shocks delivered by a stimulator (Grass Instruments Co, S48) for a period of one min (12 pulses of 90 V at a frequency of 0.2 Hz); shortly after, flies were allowed to rest for 45 s while exposed to ambient air before being exposed to a second stimulus, henceforth referred to as the Conditioned Stimulus - (CS-), for one min without experiencing any electric shocks; flies were then allowed to rest for 45 s. The conditioned stimuli were either an odor dissolved in mineral oil or mineral oil alone (Sigma-Aldrich, M5904); the odors used were farnesol (1:1000 in mineral oil; Sigma-Aldrich, 43348), hexanoic acid (1:1000 in mineral oil; Sigma-Aldrich, 21529), 3-octanol (1:1000 in mineral oil; Sigma-Aldrich, 218405) or 4-methylcyclohexanol (1:1000 in mineral oil; Sigma-Aldrich, 153095). The training phase was performed either once (single training) or repeated six times with a 15-minute-long inter-training interval (spaced training). Between training and testing, there was a resting phase during which flies were housed in regular food vials and kept in the dark. During the testing phase, flies were given the choice to enter an arm of the T-maze perfumed with CS+ or the other arm perfumed with CS-. The performance index (PI) was calculated as follows: PI = (CS+-CS-)/(CS++CS−). Each *n* reported in the data sets represents the average values obtained in a pair of reciprocal experiments; in reciprocal experiments, the stimuli used as CS+ and CS− were switched. Innate odor acuity was measured by allowing flies to choose between either a chamber perfumed with mineral oil or a chamber perfumed with an odor over the course of two minutes. Shock acuity was measured by allowing flies to choose between a chamber lined with a copper grid onto which 90 V electric shocks were delivered every five seconds over the course of one minute. All experiments were performed at 23 °C and 55–65% relative humidity under dim red light.

### Statistical analyses

For the statistical analyses of the data shown in Table 1, Figs. 3, 4, Supplementary Data 1, Supplementary Table 1 and Supplementary Fig. 3, 4 and 9, *p*-values were computed using the Mann–Whitney *U* test; statistical significance is indicated as $p < 0.05$ (*), $p < 0.01$ (**) and $p < 0.001$ (***). For the statistical analyses of the data shown in Fig. 2 and Supplementary Data 2, *p*-values were computed using the Fisher's exact test; to control for false positives, *p*-values were adjusted with a false discovery rate of 10% using a Benjamini-Hochberg procedure. For the statistical analyses of the data shown in Fig. 5, *p*-values were computed using the sample t test.

To compute the *p*-values shown in Fig. 2d, e, the connection frequencies measured in the *D. sechellia* experimental matrix were compared to the connectivity frequencies measured in a data set that contains biased shuffle matrices built using the generalist experimental matrices (*D. melanogaster* male, *D. melanogaster* female or *D. simulans*). These biased shuffle matrices preserved biases in input connection frequencies and the distribution of input counts to Kenyon cells; 1000 biased shuffle matrices were generated for each comparison. Across comparisons, the probability that the observed *D. sechellia* glomerulus connection frequency was lower (Fig. 2d) or higher (Fig. 2e) than that observed in the data set containing the biased shuffle matrices was computed.

### Reporting summary

Further information on research design is available in the Nature Portfolio Reporting Summary linked to this article.

## Data availability

The data generated in this study are provided in the Supplementary Information and Source Data file. The unprocessed data—namely, raw two-photon and confocal images—is very large (~2.5 TB). Rather than housing these data on a public repository, the corresponding author will make them available upon request. Source data are provided with this paper.

## Code availability

The codes used to analyze the data in this study are available on GitHub: https://github.com/evavigato/Ellis-et-al-2024.git.

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

## Acknowledgements

The authors thank members of the Caron laboratory for comments on the manuscript; David Stern for sharing the *attp²¹⁷⁶* and *attp²¹⁷⁸ D. simulans* strains and φC31 integrase mRNA; Vanessa Ruta for sharing the *UAS-C3PA-GFP* plasmid; Adam Lin for preparation of the standard cornmeal agar medium; the Cell Imaging Core at the University of Utah for use of the Zeiss LSM 880 microscope; Nathan Clark for general guidance generating the hierarchical clustering dendrogram and similarity trees; Ashley Platt and Miles Jacob for assistance with general laboratory concerns. This work has been funded by grants from the National Institute for Neurological Disorders and Stroke (R01 EB 029858 (A.L.K.), R01 NS 106018 (S.J.C.C.) and R01 NS 1079790 (S.J.C.C.)), the National Science Foundation (DBI 1707398 (A.L.K.) and IOS 2042397 (S.J.C.C.)), the Gatsby Charitable Foundation (A.L.K.), the European Research Council (European Research Council Advanced Grant 833548) (R.B.) and the Swiss National Science Foundation (Ambizione Grant PZOOP3 185743) (T.O.A.). Further financial support was provided by the Eunice Kennedy Shriver National Institute of Child Health and Human Development (T32-HD-007491) (K.E.E.), the Department of Energy Computational Science Graduate Fellowship (DE-SC0022158) (I.G.), the Burroughs Wellcome Foundation (A.L.K.), the McKnight Endowment Fund (A.L.K.), the Simons Collaboration on the Global Brain (A.L.K.), the Human Frontier Science Program (LT000461/2015-L) (T.O.A.) and the Georges S. and Dolores Eccles Foundation (S.J.C.C.).

## Author contributions

K.E.E., S.B., and S.J.C.C. conceived the project. K.E.E. photo-labeled projection neurons and Kenyon cells, dye-labeled individual projection neurons, and analyzed their morphology; K.E.E. generated the connectivity matrices; S.B. performed and analyzed all behavioral experiments; H.M.S. generated and analyzed the antennal lobe reconstructions; I.G. and A.L.K. performed the statistical analyses of the connectivity matrices; E.V. generated the similarity trees; T.O.A. and R.B. provided the *D. sechellia* transgenic lines and plasmids used to generate the *D. simulans* transgenes. K.E.E and S.J.C.C. wrote the manuscript with input from all other authors.

## Competing interests

The authors declare no competing interests.
