## [Peer Review File · Nature Communications]

Evolution of connectivity architecture in the *Drosophila* mushroom bodyReviewers' Comments:

Reviewer #1:

Remarks to the Author:

Ellis et al. examined the characteristics of neural connections in three species of *Drosophila* through the staining of projection neurons in the antennal lobe. First of all, it is a very large amount of work. The content is very specialized, but I think it is of great value because we do not really know what kind of interspecies differences exist in neural circuits. On the other hand, there are two points on which I disagree.

Major concerns

There are 2000 Kenyon cells, one of which is labeled, and the projection neuron that projects to it was further stained. In this experiment, not all Kenyon cells were covered, nor was there a system to randomly choose a Kenyon cell without bias. If there was a bias in the selection of Kenyon cells (as an extreme example, it would be easy to imagine a case where Kenyon cells labeled in each of the three species do not overlap at all in terms of homology, i.e., we are not comparing apples to apples), then any subsequent statistical treatment will not allow us to discuss quantitative differences in neural connections between the species. I do not know the technical details of neural connections, so if this is wrong, please refute it.

Behavioral experiments show that 1. there are interspecies differences in learning; 2. within species, an odor that activates a large number of highly connected glomeruli tends to be learned better. However, there is no evidence that the species differences in learning are due to the species-specific connection bias observed here, and perhaps no direct evidence that more connections increase learning performance. In this regard, I see no basis for the statements in lines 425, 439, and 447-449. I think it would be better to position this paper as primarily a description of anatomical data, with behavioral data as ancillary, and only statements about learning that can be said on the basis of evidence.

Minor point

In the Methods section, I think that the genotype of *D. simulans* used is not described correctly.

Reviewer #2:

Remarks to the Author:

How do the brain and behaviors evolve? This comparative study by Ellis et al tackles this question at the level of the mushroom bodies in three *Drosophila* species, *D. melanogaster*, *D. simulans*, and *D. sechellia*. Mushroom bodies are the main memory center of the *Drosophila* brain which contribute to olfactory processing and the formation of odor memories. In *Drosophila* odors are detected by olfactory receptor neuron classes, which send axons into class-specific glomeruli in the antennal lobes of the brain, where they synapse with projection neurons (PNs). In *D. melanogaster*, the connectivity map of PN inputs onto individual Kenyon cells (KC) of the mushroom body revealed that each KC receives synaptic input from a random set of projection neurons (PNs), but these connections are biased, where projection neurons from specific glomeruli are more frequently connected to KCs. In this paper, the authors take a comparative approach to describe how this connectivity map differs in the brains of 3 *Drosophila* species with known phylogenetic and ecological relationships: *D. melanogaster* (male and female), *D. simulans* (female), and *D. sechellia* (female). Among the three species, *D. melanogaster* and *D. simulans* are generalist feeders, whereas *D. sechellia* is a specialist feeder on Noni fruit endemic to the island of Seychelles. Despite these ecological and behavioral differences, *D. simulans* and *D. sechellia* are phylogenetically more similar compared to *D. melanogaster*. A comparison of the connectivity map between PNs and KCs for each glomerulus across the three species shows that the random yet biased PN-KC connections are evolutionarily conserved. But the authors detected shifts in these connectivity biases for some ecologically relevant glomerular

representations specifically in *D. sechellia*. Even though *D. sechellia* is phylogenetically more similar to *D. simulans*, the shifts in connectivity biases are higher in *D. sechellia* compared to both *D. simulans* and *D. melanogaster*. So, biases in connectivity are associated with ecology rather than phylogenetic relationships and likely contribute to behavioral shifts.

The authors also detected changes in PN numbers and bouton volume that correlate with PN-KC connectivity; an increase or decrease in PN number or PN bouton volume is associated with an increase or decrease in connectivity, respectively. Many of the species-specific differences were observed for the representation of glomeruli associated with ecologically relevant odors.

To test for behavioral consequences of shifts in connectivity biases in the mushroom bodies particularly for forming odor associations, the authors perform olfactory learning assays where they pair multiple odors, known to activate multiple glomeruli, with electric shock followed by T-maze assay with the conditioned stimulus. These showed that odors activating more connected glomeruli are more learnable across species. One interesting finding that is unexpected is that *D. sechellia*, which has increased representation of some glomeruli, is significantly more impaired in associating odors with an electric shock. The authors explain this as likely due to a defect in dopamine synthesis in *D. sechellia*. I think the correlation between connectivity biases and olfactory learning can only be reliably made for *D. simulans* and *D. melanogaster*, as there is a disproportionate amount of pseudogenization in *sechellia* compared to other species, which accompanies many changes in the metabolism of critical biomolecules and detoxification.

This study is a tremendous amount of work that morphologically compares neuroanatomical and synaptic organization in the mushroom bodies across three *Drosophila* species and provides new insights into our understanding of how olfactory circuits evolve. The data analysis is appropriate and supports the interpretations and conclusions. I have only a few comments and questions:

1- Is there a reason why PN-KC connectivity matrices from only *D. melanogaster* males are described?

2- Could the authors put the findings into the general perspective of how circuits and behaviors evolve? If a given behavior, driven by a defined circuitry, is altered at the evolutionary scale, where would neuroanatomical changes occur throughout the circuit? Do the neuroanatomical features of the whole circuitry change in a predictable way?

3- This is a relevant point to the previous one. The authors mention that species-specific shifts in connectivity biases at the mushroom body are not due to altered activity at the periphery, but likely due to variations at the population level that are selected through evolutionary bottlenecks. They claim that mushroom bodies are more integrated into multiple circuits and thus are less liable to change. However, since mushroom bodies are the odor association center, I would have predicted the opposite given the plasticity of learning behaviors and synaptic effects associated with learning. One can imagine a scenario where the feeding specializations can occur in multiple steps: First, existing population-level variation in connectivity biases helps the selection of individuals with overrepresented ecologically relevant glomerular inputs onto Kenyon cells. This in turn 2) changes the selective pressures for developmental programs favoring alterations to the number of peripheral neuronal populations tuned to those ecologically relevant odors. In this sense, it is interesting that the Or22a glomerulus (DM2) is not overrepresented in *D. sechellia* mushroom bodies, even though there is an increase in the volume of this glomerulus due to an increase in the number of ORNs. Whereas DL2d glomerular volume is increased due to an increase in the number of Ir75b neurons in *D. sechellia*. Yet in this case, there is an accompanying increase in the PN-KC connectivity bias for this glomerulus. So, it seems like some olfactory circuits show neuroanatomical changes at only specific levels of the circuit, but the ones that seem to be driving behavioral adaptations to ecological constraints are associated with neuroanatomical changes at multiple levels of the circuitry.

4- For extended figure 2. The position of the branches seems different between mel and sec/sim females for gamma; males and females in mel; and for alpha/beta between mel and sech and sim. Is this normal or does it represent connectivity/neuroanatomical differences?

5- It might be good to cite reports of species-specific analysis of developmental and transcriptional programs across multiple *Drosophila* species. Some of these reports show that transcriptional and neuroanatomical changes appear more similar between specialist species (*D. sechellia* and *D. erecta*) compared to the generalists, which do not adhere to phylogenetic relationships (i.e. Pan et al Sci Rep, 2017). The phenomenon observed here that shows some shifts correlate across ecological/behavioral dimensions, whereas others adhere to phylogenetic ones, is somewhat similar.

Reviewer #3:

Remarks to the Author:

Summary of results

The manuscript 'Evolution of connectivity architecture in the *Drosophila* mushroom body' presents an example of central neuronal circuit evolution between the antennal lobe and the mushroom bodies of three closely related but ecologically divergent *Drosophila* species (*D. melanogaster*, *D. simulans*, *D. sechellia*). In an impressive tracing effort, the authors used genetically encoded photoactivatable GFP to label individual Kenyon cells in the mushroom bodies and dye electroporation to backtrace projection neurons to examine which glomeruli in the antennal lobe connect to a given Kenyon cell. They establish that glomerular input to Kenyon cells via projection neurons is randomized in all three species but with species-specific connectivity biases. *D. sechellia* stands out, in particular, as having a distinct set of connectivity biases and there are hints that this may be related to the species' unique noni-specialist ecology. More specifically, at least one of the key glomeruli that has a higher connectivity frequency in Dsec (DL2d) is linked to preference for noni fruit volatiles. Both the number of projection neurons as well as the volume of boutons at the synapse between projection neurons and Kenyon cells can explain the observed connectivity shifts. Finally, the authors use a behavioral paradigm in which flies are learning to associate relevant food odors with punitive electric shocks and report that odors detected by glomeruli with more frequent connections to the Kenyon cells show an enhanced learning performance. Performance however varied significantly between the three examined species.

Significance

The study makes a significant advance in our understanding of neuronal circuit evolution and is one of currently very few studies reporting such changes beyond the sensory periphery. This work will thus be of interest and relevance to the field. The authors do a nice job of motivating the work and presenting relevant background information. The connectivity data are impressive and their analysis of species-specific biases generally rigorous and convincing. This said, we feel that an alternative statistical and graphical approach could substantially clarify and strengthen the conclusions about ways in which *D. sechellia* stands out from the two generalists (see below). We also had a hard time relating the behavioral experiments to the connectivity findings and found the link there unconvincing. Overall, we think the work will have a significant impact on the field and that its quality and relevance warrant publication after these issues are addressed.

Major comments:

--- This work involved a massive effort with 200+ Kenyon cells traced per species! Bravo to the team for producing this rich dataset. We found the conclusion that connectivity biases vary across species to be rigorous and convincing.

--- We also found the figures to be of high quality and mostly easy to follow. The schematics are beautiful.

--- One area of the manuscript that we found confusing/awkward was the discussion of the specific glomeruli that stand out as having different connectivity frequency in Dsec. Ultimately, the authors emphasize ways in which Dsec females differ from both Dmel and Dsim females and interpret these in light of the Dsec's unique ecology. However, the tests they use to identify these glomeruli are pairwise and speak only to whether Dsec differs from one or the other generalist. Among all the glomeruli highlighted in the paper, only one is significantly different between Dsec and both generalists (DC3). What we feel is missing is a quantitative analysis specifically aimed at identifying glomeruli that stand out in Dsec relative to both generalists.

For example, we tried summing the p-values from the Dsec-vs-Dmel and Dsec-vs-Dsim comparisons (Extended Data Table 2) to calculate the probability that the null hypothesis of equal connectivity is true in either case. We then made a new volcano plot with the $-\log P$ -value from this modified test on the Y axis and a modified fold change (comparing Dsec to the average of Dmel/Dsim) on the X axis (see separate file). The 4 most significant glomeruli from this perspective are the exact four that the authors emphasize the most in subsequent analyses (DC3, VM5v, DP1l, and DL2d). So it's not that we don't think these glomeruli are the key outliers - just that the text and figures do not convincingly show that. (Also note that some of the other glomeruli highlighted in the text are definitely not outliers from this alternative perspective).

This isn't the only way to address the data, and it's almost certainly not the most elegant. But we think that some sort of analysis designed to identify connectivity differences that set Dsec apart from both generalists would improve the manuscript and be more consistent with its pitch (ie. that these changes may be linked to Dsec's unique ecology).

--- Related to the above, the authors use trees in Figure 2 (panels e-g) to make the argument that Dsec is the outlier relative to both generalists and to identify a set of glomeruli that have similar connectivity in Dmel and Dsim despite the fact that Dsim is more closely related to Dsec phylogenetically. We found these trees awkward/misleading for a couple reasons. First, it just seemed like an overly complicated way to answer a very simple question: For which (and how many) glomeruli is Dsim more similar to Dmel than it is to its closer relative Dsec? Second, it's not clear what the outgroups represent in these trees. We don't know of any evolutionary analysis in which the character value for an outgroup is inferred or set to some default value. Moreover the presence of the outgroup doesn't add any useful information to the trees. If desired, the authors could show unrooted trees where the branch lengths reflect mean differences in connectivity among species. But we would advocate simply dropping the connectivity trees from the manuscript altogether. Instead, we think it would make sense to include a hierarchical clustering dendrogram to the left of the Shannon-Jensen matrix in Fig. 2c. It's clear from the matrix that Dsec is the outlier when looking across all glomeruli and a hierarchical clustering dendrogram would nicely highlight that. Then some sort of quantitative approach (see above) could be used to identify the individual glomeruli that drive this result.

--- One part of the paper that we found unconvincing was the behavioral analysis, which does directly support the general story of the study. Most importantly, the circuit analysis shows how individual glomeruli are more or less likely to connect to kenyon cells in some species than in others. The obvious behavioral correlate of this type of circuit difference is that learned associations are more or less easily formed with the cognate odorants of these key glomeruli in some species than in others. However, this is not what the authors see. Dsec does not learn associations with a DL2d-activated odorant (hexanoic acid) any more readily than the other two species. More generally, Dsec just doesn't seem to be a good learner. The authors note that this may reflect the specific paradigm used or a biological difference related to selection (or lack thereof) in its unique ecological niche.

The authors then pivot to focus on how well single species learn associations with different odorants —

reporting that they more easily learn associations with odorants that activate many glomeruli than with those that activate only one or a few glomeruli. This is an interesting but somewhat orthogonal result. The trend also appears to depend heavily on one odorant that activates many glomeruli and is easily learned. The authors could test additional odorants to make this result more convincing.

In summary, we suggest more cautious interpretation of the behavioral data - possibly moving it to a supplementary figure. The statement in the abstract that "behavioral analyses suggest that such increased connectivity enhances learning performance in an associated task" is not totally unjustified, but we found it misleading as the authors were not able to connect the species differences in connectivity directly to learning.

Minor comments/questions:

Line 48 and 100: Meaning of "evolutionary pressures" unclear. Do you mean simply "evolution", which would include both selection and drift? Or do you mean natural selection in particular, in which case "selection pressures" or "natural selection" might be better.

Extended Data Table 1: As stated in the intro, 3 glomeruli are known to be dramatically larger in Dsec (DL2d, DM2, and VM5d). Why don't these differences come out as significant in this analysis? Are the tests underpowered? If so, I would suggest removing them from the apper as they may mislead readers into thinking that there are no significant differences in glomerulus volume among species. (Also pls state in legend whether p-values have already been corrected for multiple testing).

Fig. 2d: Above, we suggest de-emphasizing the pairwise comparisons depicted in Fig. 2d, or at least complementing them with an additional test that looks for Dsec vs (Dmel, Dsim) differences. If the authors keep the pairwise volcano plots, we suggest adjusting the order and orientation such that (1) Dsim vs Dmel comes before the two plots contrasting Dsec to each generalist and (2) flipping the x-axis in the Dsec vs Dmel plot so that Dsec is depicted on the right hand side of both the Dsec vs Dmel and Dsec vs Dsim comparisons. This will make it easier for readers to look for glomeruli with consistent increases or decreases in connectivity in the specialist.

Discussion line 446. The authors state that this is the first evidence for evolutionary changes in the connectivity architecture of higher olfactory brain centers. What about the finding that DM2 projection neurons have different branching pattern in lateral horn of Dsec (Auer et al 2020 Nature)? Maybe we missed a subtle emphasis in the sentence that somehow excludes this example?

Line 212: We would encourage the authors to temper the conclusion that only a fraction of glomeruli account for the observed shift in biases between the three species. They show that only a fraction are significant in the pairwise tests, but whether or not differences meet a significance threshold has as much to do with the power of the tests as it does with biology. Looking at the distribution of differences might be more informative (e.g. is it continuous or is it more or less uniform with a few big outliers?)

Line 238: Why is a glomerulus that responds to a *D.melanogaster*-specific pheromone (VA1d) still above average represented in both other species? Is it known if this glomerulus has different functions in the other two species?

Fig. 3 and 4. Is there a particular reason to use presynaptic bouton volume as measure of synaptic connectivity as opposed to bouton number? Bouton number could be more intuitive and is also cited as a known reason for connectivity biases in the literature (lines 268 - 270). Number seems also a parameter that is less likely to be an imaging artifact than volume.

Related to the above, could you plot bouton volume (or number) on a per PN basis so that the value is orthogonal to PN number and better corresponds to the underlying mechanism?

Fig. 3c – Why are adPNs shown for some glomeruli and IPNs for others? Is it because those are the ones that go through MB in each set of cases?

Line 331: Could you speculate why the exact mode of how glomerular representation changes (boutons vs PN numbers) could be glomerulus specific?

Figure 2 panel d: y axis should be $-\log(p)$.

Extended Data Figure 11: Looks like data for “innate” are replotted within each panel to accompany both the Odor-as-CS and Oil-as-CS data, which makes it look like different experiments. We also found it confusing that positive performance index values sometimes applied to the odor and sometimes to oil, depending on which was the CS. Both problems could be solved by making positive index values correspond to preference for odor, regardless of whether it is the CS or US. Then the “innate” data can be plotted just once, followed by single and spaced training for odor as CS, and then single and spaced training for oil as CS.

Comments from Reviewer 1

Ellis et al. examined the characteristics of neural connections in three species of Drosophila through the staining of projection neurons in the antennal lobe. First of all, it is a very large amount of work. The content is very specialized, but I think it is of great value because we do not really know what kind of interspecies differences exist in neural circuits. On the other hand, there are two points on which I disagree.

We greatly appreciate Reviewer 1's thoughtful evaluation of our manuscript and their useful feedback. We have revised our manuscript specifically to address their feedback as well as provide answers to their questions in this document.

Major concerns:

There are 2000 Kenyon cells, one of which is labeled, and the projection neuron that projects to it was further stained. In this experiment, not all Kenyon cells were covered, nor was there a system to randomly choose a Kenyon cell without bias. If there was a bias in the selection of Kenyon cells (as an extreme example, it would be easy to imagine a case where Kenyon cells labeled in each of the three species do not overlap at all in terms of homology, i.e., we are not comparing apples to apples), then any subsequent statistical treatment will not allow us to discuss quantitative differences in neural connections between the species. I do not know the technical details of neural connections, so if this is wrong, please refute it.

Our previous work — as well as studies by others including recent analyses of two adult connectomes — have unequivocally shown that Kenyon cells receive biased-random inputs from projection neurons. This indicates that knowledge of one input to a Kenyon cell does not allow prediction of its other inputs, and that some projection neurons connect more frequently to Kenyon cells than expected under a uniform distribution of inputs while other projection neurons connect less frequently (see Caron *et al.* (2013) *Nature*, Hayashi *et al.* (2021) *Current Biology*, and Li *et al.* (2020) *Elife* for more details). We confirmed this finding in this manuscript: principal component analyses of connectivity matrices from all three species did not reveal any structural pattern in Kenyon cell inputs, apart from the observed biases, supporting the idea that connections between projection neurons and Kenyon cells are randomly assigned.

Interestingly, a study of the larval mushroom body connectome showed differences between Kenyon cells in the left and right mushroom bodies in terms of input wiring (Eichler *et al.* (2017) *Nature*). This observation not only further support the idea that Kenyon cells are randomly wired, but it also shows than Kenyon cells are not predetermined and that it is unlikely that a specific Kenyon cell can be consistently identified across mushroom bodies. Our data support this idea: we fail to identify the 'same' Kenyon cell twice. The only structure we can measure are biases. The biases we observed in the connectivity matrices are generated by the total number of connections we mapped between projection neurons and Kenyon cells in a given species. Thus, our statistical analyses can only measure these two features of Kenyon cell connectivity: random input (or lack of stereotypy) and biases. These are the features we compare across species.

Behavioral experiments show that 1. there are interspecies differences in learning; 2. within species, an odor that activates a large number of highly connected glomeruli tends to be learned better. However, there is no evidence that the species differences in learning are due to the species-specific connection bias observed here, and perhaps no direct evidence that more connections increase learning performance. In this regard, I see no basis for the statements in

lines 425, 439, and 447-449. I think it would be better to position this paper as primarily a description of anatomical data, with behavioral data as ancillary, and only statements about learning that can be said on the basis of evidence.

We agree that our main findings are based on anatomical evidence, and that our behavioral studies do not provide direct evidence that connectivity frequencies lead to better learning performance. Consequently, we have revised the sections highlighted by Reviewer 1:

Lines 422-423: "Our study also suggests that increased connectivity might enhance learning performance in an associative task."

Lines 436-438: "The fact that we were able to correlate a visible neuronal phenotype with these connectivity changes and correlate connectivity frequencies with learning performance, makes such investigations theoretically possible in the future. "

Lines 456-458: "Our study provides the first evidence at the cellular ~~and behavioral~~ level for such evolutionary changes in the connectivity architecture of higher olfactory brain centers and how these changes might enhance cognitive functions."

We have also tempered our wording in other sections to ensure clarity and accuracy, employing more precise language to avoid any misleading interpretations.

Minor point

In the Methods section, I think that the genotype of D. simulans used is not described correctly.

We thank Reviewer 1 for this comment. We have now added an additional section in Methods describing how the *D. simulans* $y^1w^1;pBac\{3XP3::EYFP, N-synaptobrevin-GAL4\}^{attp2176};;$ and $y^1w^1;;pBac\{3XP3::EYFP, UAS-C3PA-GFP\}^{attp2178}$ transgenic lines were generated.

Lines 918-929:

"Transgenesis

A *N-synaptobrevin-GAL4* (*D. simulans*) plasmid was generated by cloning a 1.9 kb sequence upstream of *N-synaptobrevin* using PCR and genomic DNA extracted from *D. simulans attp²¹⁷⁶* flies (forward oligonucleotide: GATCGGTACCGAACTCGTCCTCAAAGATGGAAACAGAG; reverse oligonucleotide: GATCGCGGCCGCGAATTCGGCTGGCGATGATTAGGATG); the amplified sequence was inserted into the pGal4attB plasmid using the NotI and KpnI restriction sites; this *N-synaptobrevin-GAL4* (*D. simulans*) plasmid was injected into the *D. simulans attp²¹⁷⁶* strain with the ϕ C31 integrase following a standard protocol (BestGene) resulting in the $y^1w^1;pBac\{3XP3::EYFP, N-synaptobrevin-GAL4\}^{attp2176};;$ transgenic line⁴³. The $y^1w^1;;pBac\{3XP3::EYFP, UAS-C3PA-GFP\}^{attp2178}$ transgenic line was generated by injecting a *UAS-C3PA-GFP* plasmid into the *D. simulans attp²¹⁷⁸* strain using similar protocols⁴⁴."

Comments from Reviewer 2

How do the brain and behaviors evolve? This comparative study by Ellis et al tackles this question at the level of the mushroom bodies in three Drosophila species, D. melanogaster, D. simulans, and D. sechellia. Mushroom bodies are the main memory center of the Drosophila brain which

contribute to olfactory processing and the formation of odor memories. In Drosophila odors are detected by olfactory receptor neuron classes, which send axons into class-specific glomeruli in the antennal lobes of the brain, where they synapse with projection neurons (PNs). In D. melanogaster, the connectivity map of PN inputs onto individual Kenyon cells (KC) of the mushroom body revealed that each KC receives synaptic input from a random set of projection neurons (PNs), but these connections are biased, where projection neurons from specific glomeruli are more frequently connected to KCs. In this paper, the authors take a comparative approach to describe how this connectivity map differs in the brains of 3 Drosophila species with known phylogenetic and ecological relationships: D. melanogaster (male and female), D. simulans (female), and D. sechellia (female). Among the three species, D. melanogaster and D. simulans are generalist feeders, whereas D. sechellia is a specialist feeder on Noni fruit endemic to the island of Seychelles. Despite these ecological and behavioral differences, D. simulans and D. sechellia are phylogenetically more similar compared to D. melanogaster. A comparison of the connectivity map between PNs and KCs for each glomerulus across the three species shows that the random yet biased PN-KC connections are evolutionarily conserved. But the authors detected shifts in these connectivity biases for some ecologically relevant glomerular representations specifically in D. sechellia. Even though D. sechellia is phylogenetically more similar to D. simulans, the shifts in connectivity biases are higher in D. sechellia compared to both D. simulans and D. melanogaster. So, biases in connectivity are associated with ecology rather than phylogenetic relationships and likely contribute to behavioral shifts. The authors also detected changes in PN numbers and bouton volume that correlate with PN-KC connectivity; an increase or decrease in PN number or PN bouton volume is associated with an increase or decrease in connectivity, respectively. Many of the species-specific differences were observed for the representation of glomeruli associated with ecologically relevant odors. To test for behavioral consequences of shifts in connectivity biases in the mushroom bodies particularly for forming odor associations, the authors perform olfactory learning assays where they pair multiple odors, known to activate multiple glomeruli, with electric shock followed by T-maze assay with the conditioned stimulus. These showed that odors activating more connected glomeruli are more learnable across species. One interesting finding that is unexpected is that D. sechellia, which has increased representation of some glomeruli, is significantly more impaired in associating odors with an electric shock. The authors explain this as likely due to a defect in dopamine synthesis in D. sechellia. I think the correlation between connectivity biases and olfactory learning can only be reliably made for D. simulans and D. melanogaster, as there is a disproportionate amount of pseudogenization in D. sechellia compared to other species, which accompanies many changes in the metabolism of critical biomolecules and detoxification.

This study is a tremendous amount of work that morphologically compares neuroanatomical and synaptic organization in the mushroom bodies across three Drosophila species and provides new insights into our understanding of how olfactory circuits evolve. The data analysis is appropriate and supports the interpretations and conclusions. I have only a few comments and questions:

We thank Reviewer 2 for their insightful feedback on our manuscript. We are grateful for the recognition of the extensive work involved in our comparative analysis of the neuroanatomical and synaptic organization in the mushroom body architecture across *Drosophila* species. The constructive comments and questions from Reviewer 2 have been invaluable in refining our manuscript. We have carefully considered and incorporated these suggestions, making necessary modifications accordingly.

1- Is there a reason why PN-KC connectivity matrices from only D. melanogaster males are described?

The selection of *D. melanogaster* male connectivity matrices for our study was driven by specific methodological considerations. The transgenes essential for our mapping technique, *nSynaptobrevin-GAL4* and *UAS-photoactivatable-GFP*, are both located on the X chromosome in *D. sechellia*, limiting our analysis to female flies. Initially, our reference was the male *D. melanogaster* connectivity matrix from Caron *et al.* (2013) Nature. To ensure consistency and comparability, we replicated this matrix for male *D. melanogaster* in our study. Additionally, to assess any potential sexual dimorphism in connectivity biases, we compared matrices between male and female *D. melanogaster* and found no meaningful differences. Given these findings, we did not deem it necessary to extend this aspect of our study to *D. simulans*, as there was no strong indication of sexually dimorphic biases that would impact our research goals. We think this is valuable information and now include it in Methods (see Lines 1005-1013).

2- Could the authors put the findings into the general perspective of how circuits and behaviors evolve? If a given behavior, driven by a defined circuitry, is altered at the evolutionary scale, where would neuroanatomical changes occur throughout the circuit? Do the neuroanatomical features of the whole circuitry change in a predictable way?

In response to this feedback, we significantly revised the Discussion section. Specifically, we have refocused the third paragraph to address the inquiries posed by Reviewer 2, offering our insights and positioning our study within the broader context of existing knowledge:

Lines 443-458: "Despite these limitations, our study contributes new insights into the emergence of behavioral adaptations. Behavioral adaptations arise from modifications in the way neuronal circuits process information, and it is known that different cellular mechanisms can give rise to such modifications^{36,37}. For instance, changes in the expression levels of receptors or ion channels can alter the electrophysiological properties of neurons and, consequently, neuronal output. There are many documented examples of such evolutionary changes at the periphery, including in the *Drosophila* olfactory systems, showing that sensory systems can adapt by finely tuning their detection capabilities to features of the environment peculiar to a species³⁸. Changes in neuronal connectivity, whether through changes in synaptic weights between existing partners or through the formation of new synaptic partners, can also alter the way information flows in a circuit. In principle, such modifications can occur at any level within a circuit. However — perhaps because neurons at the periphery are more accessible than neurons embedded in the higher brain centers — there is a predominance of documented functional changes in sensory neurons over connectivity changes in central nervous systems. Our study provides the first evidence at the cellular level for such evolutionary changes in the connectivity architecture of higher olfactory brain centers and how these changes might enhance cognitive functions."

3- This is a relevant point to the previous one. The authors mention that species-specific shifts in connectivity biases at the mushroom body are not due to altered activity at the periphery, but likely due to variations at the population level that are selected through evolutionary bottlenecks. They claim that mushroom bodies are more integrated into multiple circuits and thus are less liable to change. However, since mushroom bodies are the odor association center, I would have predicted the opposite given the given the plasticity of learning behaviors and synaptic effects associated with learning. One can imagine a scenario where the feeding specializations can occur in multiple steps: First, existing population-level variation in connectivity biases helps the selection of individuals with overrepresented ecologically relevant glomerular inputs onto Kenyon cells. This in turn 2) changes the selective pressures for developmental programs favoring alterations to the number of peripheral neuronal populations tuned to those ecologically relevant odors. In this sense, it is interesting that the Or22a glomerulus (DM2) is not overrepresented in D. sechellia

mushroom bodies, even though there is an increase in the volume of this glomerulus due to an increase in the number of ORNs. Whereas DL2d glomerular volume is increased due to an increase in the number of Ir75b neurons in D. sechellia. Yet in this case, there is an accompanying increase in the PN-KC connectivity bias for this glomerulus. So, it seems like some olfactory circuits show neuroanatomical changes at only specific levels of the circuit, but the ones that seem to be driving behavioral adaptations to ecological constraints are associated with neuroanatomical changes at multiple levels of the circuitry.

We appreciate Reviewer 2's insightful observation, which indeed raises fundamental questions central to our research program. The interplay between neuroanatomical changes and behavioral adaptations, particularly in response to ecological constraints, is a complex problem. The reviewer's hypothesis about the stepwise evolution, starting from population-level variation in connectivity biases to selective pressures shaping developmental programs, presents a compelling framework for understanding evolutionary adaptations. The specific examples cited by the reviewer, such as the variation in the Or22a and Ir75b neurons and their associated glomeruli in *D. sechellia*, underscore the nuanced and layer-specific nature of the neuroanatomical changes necessary to drive such adaptations.

While we agree that these observations are more reflective of a broader commentary than a direct question, they indeed highlight areas for future investigation. The extent to which certain neuroanatomical changes are more adaptive than others, and the mechanisms that confer such adaptiveness, are intriguing questions. Addressing these would require a more in-depth analysis of the anatomical changes and their direct links to behavioral adaptations, which forms a key focus of our ongoing research endeavors in the Caron, Auer and Benton laboratories. We believe that further exploration in this direction will significantly advance our understanding of the evolutionary dynamics of sensory systems.

4- For extended figure 2. The position of the branches seems different between mel and sec/sim females for gamma; males and females in mel; and for alpha/beta between mel and sech and sim. Is this normal or does it represent connectivity/neuroanatomical differences?

The variability observed in Extended Data Figure 2 is normal and expected. The morphology of Kenyon cells varies greatly regardless of their type. Depending on the location of the cell body of a Kenyon cell, it will extend its axon either straight through the calyx (for instance, the α/β Kenyon cell shown in male *D. melanogaster* and *D. sechellia*) or at the edges of the calyx (for instance, the α/β Kenyon cell shown in female *D. melanogaster* and *D. simulans*). Similarly, the position of the branches varies among Kenyon cells, even those with similar axonal paths. We decided to show this variability in the examples selected in Extended Data Figure 2. We did not measure and compare branch length in these analyses because it ultimately does not influence the connectivity architecture of the mushroom body; the number of claws per Kenyon cells would have an influence, however, and this is the number we measured. The number of claws per Kenyon cell does not vary across species.

*5- It might be good to cite reports of species-specific analysis of developmental and transcriptional programs across multiple Drosophila species. Some of these reports show that transcriptional and neuroanatomical changes appear more similar between specialist species (*D. sechellia* and *D. erecta*) compared to the generalists, which do not adhere to phylogenetic relationships (i.e. Pan et al Sci Rep, 2017). The phenomenon observed here that shows some shifts correlate across ecological/behavioral dimensions, whereas others adhere to phylogenetic ones, is somewhat similar.*

Reviewer 2 raises a good point, and we are now referring to the Pan et al. (2017) study in the Discussion. We are sorry for this omission.

Line 444-446: "Behavioral adaptations arise from modifications in the way neuronal circuits process information, and it is known that different cellular mechanisms can give rise to such modifications³⁶⁻³⁸."

Comments from Reviewer 3.

Summary of results

The manuscript 'Evolution of connectivity architecture in the Drosophila mushroom body' presents an example of central neuronal circuit evolution between the antennal lobe and the mushroom bodies of three closely related but ecologically divergent Drosophila species (D. melanogaster, D. simulans, D. sechellia). In an impressive tracing effort, the authors used genetically encoded photoactivatable GFP to label individual Kenyon cells in the mushroom bodies and dye electroporation to back trace projection neurons to examine which glomeruli in the antennal lobe connect to a given Kenyon cell. They establish that glomerular input to Kenyon cells via projection neurons is randomized in all three species but with species-specific connectivity biases. D. sechellia stands out, in particular, as having a distinct set of connectivity biases and there are hints that this may be related to the species' unique noni-specialist ecology. More specifically, at least one of the key glomeruli that has a higher connectivity frequency in Dsec (DL2d) is linked to preference for noni fruit volatiles. Both the number of projection neurons as well as the volume of boutons at the synapse between projection neurons and Kenyon cells can explain the observed connectivity shifts. Finally, the authors use a behavioral paradigm in which flies are learning to associate relevant food odors with punitive electric shocks and report that odors detected by glomeruli with more frequent connections to the Kenyon cells show an enhanced learning performance. Performance however varied significantly between the three examined species.

Significance

The study makes a significant advance in our understanding of neuronal circuit evolution and is one of currently very few studies reporting such changes beyond the sensory periphery. This work will thus be of interest and relevance to the field. The authors do a nice job of motivating the work and presenting relevant background information. The connectivity data are impressive and their analysis of species-specific biases generally rigorous and convincing. This said, we feel that an alternative statistical and graphical approach could substantially clarify and strengthen the conclusions about ways in which D. sechellia stands out from the two generalists (see below). We also had a hard time relating the behavioral experiments to the connectivity findings and found the link there unconvincing. Overall, we think the work will have a significant impact on the field and that its quality and relevance warrant publication after these issues are addressed.

We are thankful to Reviewer 3 for their positive feedback and valuable suggestions. The recognition of our study's impact in the field of neuronal circuit evolution is greatly appreciated. We took note of the recommendation for an alternative statistical and graphical approach, especially regarding *D. sechellia*, and we revised our manuscript accordingly. We also acknowledge the concerns about the linkage between our behavioral experiments and connectivity findings — a point echoed by Reviewer 1 — and we addressed this to ensure clarity and strength in our revised manuscript. Incorporating these changes has considerably enhanced the clarity of our manuscript and we are thankful for that.

Major comments:

This work involved a massive effort with 200+ Kenyon cells traced per species! Bravo to the team for producing this rich dataset. We found the conclusion that connectivity biases vary across species to be rigorous and convincing.

We also found the figures to be of high quality and mostly easy to follow. The schematics are beautiful.

One area of the manuscript that we found confusing/awkward was the discussion of the specific glomeruli that stand out as having different connectivity frequency in Dsec. Ultimately, the authors emphasize ways in which Dsec females differ from both Dmel and Dsim females and interpret these in light of the Dsec's unique ecology. However, the tests they use to identify these glomeruli are pairwise and speak only to whether Dsec differs from one or the other generalist. Among all the glomeruli highlighted in the paper, only one is significantly different between Dsec and both generalists (DC3). What we feel is missing is a quantitative analysis specifically aimed at identifying glomeruli that stand out in Dsec relative to both generalists.

For example, we tried summing the p-values from the Dsec-vs-Dmel and Dsec-vs-Dsim comparisons (Extended Data Table 2) to calculate the probability that the null hypothesis of equal connectivity is true in either case. We then made a new volcano plot with the $-\log P$ -value from this modified test on the Y axis and a modified fold change (comparing Dsec to the average of Dmel/Dsim) on the X axis (see separate file). The 4 most significant glomeruli from this perspective are the exact four that the authors emphasize the most in subsequent analyses (DC3, VM5v, DP1I, and DL2d). So it's not that we don't think these glomeruli are the key outliers - just that the text and figures do not convincingly show that. (Also note that some of the other glomeruli highlighted in the text are definitely not outliers from this alternative perspective). This isn't the only way to address the data, and it's almost certainly not the most elegant. But we think that some sort of analysis designed to identify connectivity differences that set Dsec apart from both generalists would improve the manuscript and be more consistent with its pitch (ie. that these changes may be linked to Dsec's unique ecology).

We appreciate the suggestion from Reviewer 3, which prompted us to refine our description of the statistical analyses we used to analyze shifts in biases. In response to this feedback, we developed a new statistical method for comparing *D. sechellia* with the generalists, aiming to achieve results similar to theirs. Specifically, we performed pairwise comparison using the connectivity frequencies measured in the *D. sechellia* matrix with the frequencies obtained using a set of biased shuffle matrices generated from using the generalist matrices. Notably, and as Reviewer 3 predicted, the four glomeruli central to our study — DL2d, DP1I, DC3, and VM5v — were highlighted in this analysis. Consequently, we have made significant revisions to the 'Biases in connectivity correlate with the chemical ecology of a species' section (Lines 165-252) and to Figure 2 to include this analysis and to remove parts of the text that were too long and confusing.

Related to the above, the authors use trees in Figure 2 (panels e-g) to make the argument that Dsec is the outlier relative to both generalists and to identify a set of glomeruli that have similar connectivity in Dmel and Dsim despite the fact that Dsim is more closely related to Dsec phylogenetically. We found these trees awkward/misleading for a couple reasons. First, it just seemed like an overly complicated way to answer a very simple question: For which (and how many) glomeruli is Dsim more similar to Dmel than it is to its closer relative Dsec? Second, it's not clear what the outgroups represent in these trees. We don't know of any evolutionary analysis in which the character value for an outgroup is inferred or set to some default value. Moreover,

the presence of the outgroup doesn't add any useful information to the trees. If desired, the authors could show unrooted trees where the branch lengths reflect mean differences in connectivity among species. But we would advocate simply dropping the connectivity trees from the manuscript altogether. Instead, we think it would make sense to include a hierarchical clustering dendrogram to the left of the Shannon-Jensen matrix in Fig. 2c. It's clear from the matrix that Dsec is the outlier when looking across all glomeruli and a hierarchical clustering dendrogram would nicely highlight that. Then some sort of quantitative approach (see above) could be used to identify the individual glomeruli that drive this result.

This is another great suggestion, and we revised the manuscript accordingly by replacing the phylogenetic trees with a hierarchical clustering dendrogram (Figure 2g; Lines 243-252).

One part of the paper that we found unconvincing was the behavioral analysis, which does directly support the general story of the study. Most importantly, the circuit analysis shows how individual glomeruli are more or less likely to connect to kenyon cells in some species than in others. The obvious behavioral correlate of this type of circuit difference is that learned associations are more or less easily formed with the cognate odorants of these key glomeruli in some species than in others. However, this is not what the authors see. Dsec does not learn associations with a DL2d-activated odorant (hexanoic acid) any more readily than the other two species. More generally, Dsec just doesn't seem to be a good learner. The authors note that this may reflect the specific paradigm used or a biological difference related to selection (or lack thereof) in its unique ecological niche.

The authors then pivot to focus on how well single species learn associations with different odorants — reporting that they more easily learn associations with odorants that activate many glomeruli than with those that activate only one or a few glomeruli. This is an interesting but somewhat orthogonal result. The trend also appears to depend heavily on one odorant that activates many glomeruli and is easily learned. The authors could test additional odorants to make this result more convincing.

In summary, we suggest more cautious interpretation of the behavioral data - possibly moving it to a supplementary figure. The statement in the abstract that "behavioral analyses suggest that such increased connectivity enhances learning performance in an associated task" is not totally unjustified, but we found it misleading as the authors were not able to connect the species differences in connectivity directly to learning.

We appreciate Reviewer 3's input, a sentiment shared by Reviewer 1, on the need for cautious interpretation of our behavioral results. While we understand the suggestion to move this figure to the Extended Data, we respectfully maintain our position that the behavioral analysis is crucial to the main text. This data set is particularly significant given the scarcity of studies on conditioning in these species and the comparison of their learning performances.

In response to Reviewer 3's suggestion, we have expanded our analysis by testing two additional odors, 2,3-butanedione and isopentyl acetate, selected based on the connectivity frequencies of the glomeruli they activate. These odors, we estimated, would have a cumulative connectivity frequency of about 22% in *D. sechellia* and 16 to 25% in the generalists (see Extended Data Table 4). Our results show that all three species, including *D. sechellia*, can learn these odors, a result that strengthens our conclusions. This new data set is now included as Extended Data Figure 11.

We acknowledge that these findings, while suggestive, do not conclusively prove that increased connectivity between projection neurons and Kenyon cells leads to enhanced learning

performance; our evidence remains correlative. We have revised our interpretation of these results and tempered our claims in the section now entitled "Connectivity frequencies and learning performance" (lines 343-408; also, see response to Reviewer 1). While testing additional odors or extending these analyses to the spaced training protocol would be significant undertakings, the Caron laboratory is actively pursuing related research. In unpublished studies, we are directly addressing these questions in *D. melanogaster*, and preliminary findings suggest that connectivity biases facilitate learning (MacKenzie et al., in preparation).

We hope that Reviewer 3 will appreciate the extent of our efforts to address their concerns and understand our position on the feasibility of conducting additional experiments at this stage.

Minor comments/questions:

Line 48 and 100: Meaning of "evolutionary pressures" unclear. Do you mean simply "evolution", which would include both selection and drift? Or do you mean natural selection in particular, in which case "selection pressures" or "natural selection" might be better.

We agree with Reviewer 3 and we replaced "evolutionary pressure" with "selection pressures" in both cases.

Extended Data Table 1: As stated in the intro, 3 glomeruli are known to be dramatically larger in Dsec (DL2d, DM2, and VM5d). Why don't these differences come out as significant in this analysis? Are the tests underpowered? If so, I would suggest removing them from the appear as they may mislead readers into thinking that there are no significant differences in glomerulus volume among species. (Also pls state in legend whether p-values have already been corrected for multiple testing).

In response to Reviewer 3's concern about the absence of significant differences for the DL2d, DM2, and VM5d glomeruli in Extended Data Table 1, we acknowledge the issue of underpowered tests, as we reconstructed only three antennal lobes per species. Additionally, variability in measuring neuropil volumes in fixed tissues contributed to this outcome. Although the glomerular volumes in *D. sechellia* were larger, they were not statistically significant within our sample size. These results were included for their relevance in confirming the presence of all 51 glomeruli across species for our connectivity architecture analysis. To prevent confusion, we have removed the *p*-values from Extended Data Table 1 in the revised manuscript.

Fig. 2d: Above, we suggest de-emphasizing the pairwise comparisons depicted in Fig. 2d, or at least complementing them with an additional test that looks for Dsec vs (Dmel, Dsim) differences. If the authors keep the pairwise volcano plots, we suggest adjusting the order and orientation such that (1) Dsim vs Dmel comes before the two plots contrasting Dsec to each generalist and (2) flipping the x-axis in the Dsec vs Dmel plot so that Dsec is depicted on the right hand side of both the Dsec vs Dmel and Dsec vs Dsim comparisons. This will make it easier for readers to look for glomeruli with consistent increases or decreases in connectivity in the specialist.

We agree with Reviewer 3, and figure 2 now includes the new analysis comparing *D. sechellia* with the generalists as discussed above. We have also changed the order and orientation of the volcano plots so that *D. sechellia* is always depicted on the left-hand side of the plots as suggested.

Discussion line 446. The authors state that this is the first evidence for evolutionary changes in the connectivity architecture of higher olfactory brain centers. What about the finding that DM2

projection neurons have different branching pattern in lateral horn of *Dsec* (Auer et al 2020 Nature)? Maybe we missed a subtle emphasis in the sentence that somehow excludes this example?

While Auer *et al.* (2020) indeed reported a species-specific branching pattern in *D. sechellia*, this observation primarily highlighted anatomical variations of homologous neuron across species rather than direct evidence of changes in connectivity. The distinction lies in the approach and depth of analysis regarding connectivity architecture. Our current study advances beyond mere anatomical observations by employing a technique that not only identifies but also quantitatively and qualitatively measures connectivity patterns. This method, first detailed in Caron *et al.* (2013), has been experimentally validated for mapping functional connections between neuron types. It allows us to explore the global changes in the connectivity architecture of a higher brain center — the mushroom body — across different species. While Auer *et al.* (2020) provided valuable insights into morphological differences of individual projection neurons, our study is unique in its comprehensive approach to analyzing and comparing connectivity patterns on a broader scale within Kenyon cells. To the best of our knowledge, this level of detailed connectivity analysis in higher brain centers, particularly in the context of evolutionary changes, has not been achieved in previous studies including our own, Auer *et al.* (2020).

Line 212: We would encourage the authors to temper the conclusion that only a fraction of glomeruli account for the observed shift in biases between the three species. They show that only a fraction are significant in the pairwise tests, but whether or not differences meet a significance threshold has as much to do with the power of the tests as it does with biology. Looking at the distribution of differences might be more informative (e.g. is it continuous or is it more or less uniform with a few big outliers?)

We agree with Reviewer 3's suggestion and have significantly revised the section of the results entitled "Biases in connectivity correlate with the chemical ecology of a species" (Lines 165-253) to include the new analysis discussed above. Now, our focus is on the four glomeruli that show the most notable shifts when contrasting *D. sechellia* with the generalists, directly addressing one of the major points raised by Reviewer 3.

Line 238: Why is a glomerulus that responds to a D.melanogaster-specific pheromone (VA1d) still above average represented in both other species? Is it known if this glomerulus has different functions in the other two species?

We have removed this part of the text to include the new analysis.

Fig. 3 and 4. Is there a particular reason to use presynaptic bouton volume as measure of synaptic connectivity as opposed to bouton number? Bouton number could be more intuitive and is also cited as a known reason for connectivity biases in the literature (lines 268 - 270). Number seems also a parameter that is less likely to be an imaging artifact than volume.

We agree with Reviewer 3 that, in theory, bouton number is a simpler anatomical feature to understand than bouton volume. However, identifying individual boutons becomes challenging in our samples, particularly when projection neurons are photo-labeled. This issue is amplified in cases where boutons are clustered and multiple neurons are photo-labeled, making it difficult to discern individual boutons without specific molecular markers. It is important to note that this challenge does not arise when individual neurons are dye-labeled, as it is possible to distinguish individual boutons in these samples. Therefore, due to these constraints, we opted to measure 'bouton volume' — the total volume of the boutons from a specific type of projection neuron —

rather than attempting to count each bouton in samples where multiple projection neurons are photo-labeled.

Related to the above, could you plot bouton volume (or number) on a per PN basis so that the value is orthogonal to PN number and better corresponds to the underlying mechanism?

The bouton volume reported in Figure 3 was measured for individual photo-labeled projection neurons; the bouton volume reported in Figure 4 was measured for individual dye-labeled projection neurons. We are not confident that the volumes measured by the two methods can be compared at this point and therefore we prefer not to make such a statement.

Fig. 3c – Why are adPNs shown for some glomeruli and IPNs for others? Is it because those are the ones that go through MB in each set of cases?

Glomeruli are either innervated by adPNs or IPNs, but never by both types of neuron.

Line 331: Could you speculate why the exact mode of how glomerular representation changes (boutons vs PN numbers) could be glomerulus specific?

We added a sentence to our manuscript.

Lines 324-327: "These observations further suggest that selection pressures may influence shifts in glomerular representation through at least two different types of molecular mechanism — those regulating synaptogenesis and those regulating neurogenesis — and that these mechanisms could be glomerular specific."

Figure 2 panel d: y axis should be -log(p).

We fixed this error in Figure 2f (formerly Figure 2d).

Extended Data Figure 11: Looks like data for “innate” are replotted within each panel to accompany both the Odor-as-CS and Oil-as-CS data, which makes it look like different experiments. We also found it confusing that positive performance index values sometimes applied to the odor and sometimes to oil, depending on which was the CS. Both problems could be solved by making positive index values correspond to preference for odor, regardless of whether it is the CS or US. Then the “innate” data can be plotted just once, followed by single and spaced training for odor as CS, and then single and spaced training for oil as CS.

Following a well-established convention in the field, the performance index (PI) is calculated using a standard formula:

$$PI = \frac{(CS^+ - CS^-)}{(CS^+ + CS^-)}$$

where CS^+ represents the number of flies that selected the conditioned stimulus and CS^- the number of flies that selected the unconditioned stimulus. Positive values thus indicate attraction towards the conditioned stimulus, whereas negative values indicate repulsion. This formula is a widely employed method, particularly for studies employing the T-maze assay, where results are routinely reported using Performance Indices. In our Extended Data Figure 9 (previously

Extended Data Figure 11), we show the results obtained for the reciprocal experiments in which CS⁺ alternates between an odor and oil, depending on the setup.

Reviewer 3 correctly observed that the "innate" data sets is replicated in both experiments, but the sign of the Preference Index changes based on whether the odor is the CS⁺ or the CS⁻. We believe that adhering to Reviewer 3's suggestion to modify this figure could lead to more confusion. Such a change might obscure the fact that two distinct experimental setups are used and that the formula has been correctly applied in each context. Our current presentation aims to maintain clarity, accurately reflect the different experimental conditions and follow the conventions of the field. We hope that Reviewer 3 will appreciate these considerations.

Reviewers' Comments:

Reviewer #1:

Remarks to the Author:

The issues I raised have been well addressed.

Reviewer #2:

Remarks to the Author:

The authors have addressed my concerns and I am pleased with the current revised version of the manuscript. It reflects a significant effort and offers a crucial evolutionary comparison of the connectivity features between the mushroom body Kenyon cells and olfactory projection neurons in *Drosophila* species with known behavioral feeding specializations.

Reviewer #3:

None